# LiveMCP-101: Stress Testing and Diagnosing MCP-enabled Agents on Challenging Queries

## Abstract

Tool calling has emerged as a critical capability for AI agents to interact with the real world and solve complex tasks. In contrast to conventional tool calling frameworks that rely on static, provider-specific tool definitions, the Model Context Protocol (MCP) offers a unified interface enabling agents to discover and invoke tools dynamically. However, there is a significant gap in benchmarking how well agents can solve multi-step tasks using diverse MCP tools in realistic, dynamic scenarios. In this work, we present LiveMCP-101, a benchmark of 101 carefully curated real-world queries, refined through iterative LLM rewriting and manual revision, that require coordinated use of multiple MCP tools. To address temporal variability in real-world tool responses, we introduce a parallel evaluation framework where a reference agent executes a validated plan simultaneously with the evaluated agent to produce real-time reference outputs, rather than relying on static ground-truth answers. Experiments show that even frontier LLMs achieve a task success rate below 60%, highlighting major challenges in multi-step tool use. Comprehensive error analysis identifies seven failure modes spanning tool planning, parameterization, and output handling, pointing to concrete directions for improving current models. LiveMCP-101 sets a rigorous standard for evaluating real-world agent capabilities, advancing toward autonomous agent systems that reliably execute complex tasks through MCP tool orchestration.

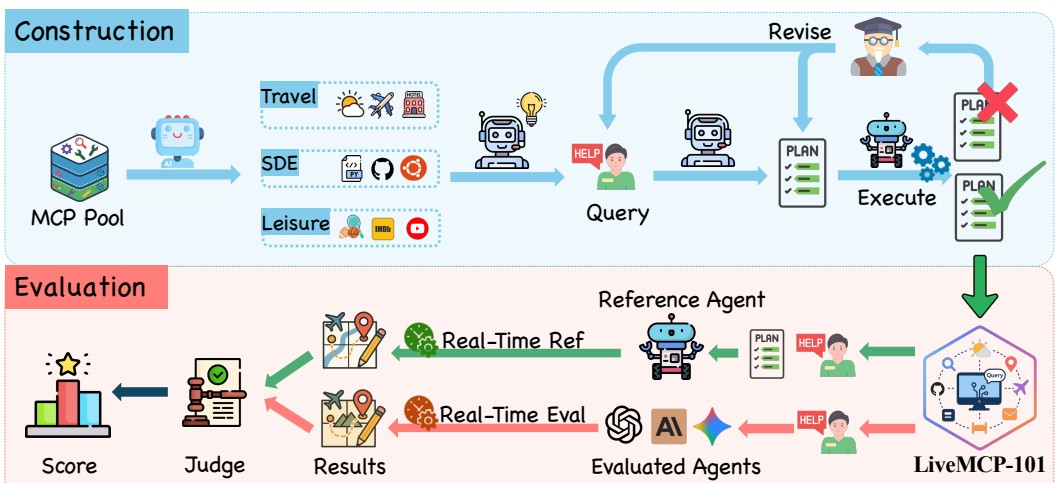

Figure 1: Construction and evaluation framework of LiveMCP-101. **Construction (top):** Queries and their corresponding execution plans are generated, refined, and validated through an iterative loop of execution, LLM-assisted editing, and manual revision. **Evaluation (bottom):** A reference agent (following the validated plan) and an evaluated agent are executed in parallel, enabling an LLM judge to score the evaluated agent by comparing their co-temporal real-time outputs.

# 1 INTRODUCTION

The ability to interact with external tools and services is a cornerstone of autonomous AI agents (Schick et al., 2023; Qin et al., 2023a), enabling them to extend their capabilities beyond static knowledge and engage dynamically with the real world. Traditional tool calling frameworks require static, provider-specific tool definitions (Li et al., 2023b; Patil et al., 2024), limiting scalability and interoperability. Model Context Protocol (Anthropic, 2024) addresses this by providing a standardized protocol that enables dynamic tool discovery, where agents query available tools from any MCP server. These developments promise a new generation of AI agents capable of executing complex, multi-step tasks with minimal human intervention (Yao et al., 2022; Shinn et al., 2024). However, reliability remains a key barrier to real-world deployment, as systems that perform well in controlled settings often fail on diverse user queries and in real production environments (Lu et al., 2024; Yao et al., 2024; Barres et al., 2025).

Understanding why agents fail (Zhang et al., 2025a; Cemri et al., 2025) in realistic, temporally evolving environments is essential for improving the corresponding models and system architectures. However, existing benchmarks (Li et al., 2023a; Tang et al., 2023; Xu et al., 2023; Patil et al., 2024; Liu et al., 2025) primarily target single-step tool calls, synthetic or static environments, or narrow tool sets. This makes them inadequate for evaluating MCP-enabled agents, which rely on dynamically discovering and coordinating tools across diverse, real-time backends. In practice, agents face live tools whose responses may vary over time and span disparate domains. Moreover, user queries often include nuanced context and strict constraints (Zhong et al., 2025), requiring accurate multi-step reasoning and tool orchestration. Consequently, existing benchmarks cannot fully reveal the gaps in current agent systems in real-world deployments.

To address these challenges and rigorously stress-test frontier LLM agents in realistic, challenging scenarios, we introduce **LiveMCP-101**, a benchmark of 101 diverse, real-world multi-step tasks that require coordinated use of MCP tools. The benchmark spans 41 MCP servers and 260 tools and is stratified into three difficulty tiers. Queries are refined through iterative LLM rewriting with human-in-the-loop revision (Wang et al., 2022) to ensure clarity, balanced difficulty, solvability with the provided tools, and objective verifiability. Each query is paired with a validated execution plan that specifies an MCP tool sequence with explicit parameterization and post-processing code to extract key information from tool-call outputs. The plans involve an average of 5.4 tool calls, with a maximum of 15. To handle time-varying tool responses, we adopt a parallel evaluation anchored in these plans. A reference agent strictly follows the plan to produce a co-temporal reference output, while the evaluated agent autonomously plans and executes. An LLM judge then scores the evaluated agents final outputs and execution trajectories against the reference.

Experiments across 18 representative LLMs show that even top-performing models (e.g., GPT-5) achieve a task success rate below 60%, revealing a substantial gap between current agent capabilities and the demands of real-world task execution. Our detailed analysis of agent trajectories (Chen et al., 2023) identifies seven common failure modes spanning tool planning and orchestration, parameterization, and output handling. We also observe that closed-source models exhibit a log-shaped trend in token efficiency, improving rapidly with small token budgets before plateauing. In contrast, open-source models show lower efficiency in converting tokens into reliable evidence.

In summary, our contributions are as follows:

- We introduce **LiveMCP-101**, a benchmark of 101 real-world, multi-step tasks for evaluating coordinated MCP tool use in dynamic environments.

- We propose a **parallel evaluation** approach anchored in validated execution plans to account for the evolving nature of real-world scenarios.

- We conduct a comprehensive evaluation across 18 models, results show that even frontier LLMs attain a task success rate under **60%**, underscoring major challenges for MCP-enabled agents in real-world multi-step tool use.

- We provide a detailed failure analysis on representative models, identifying three primary error categories: **tool planning and orchestration errors**, **parameter errors**, and **output handling errors** with seven subtypes, informing targeted improvements to enhance agent capability in dynamic, multi-step MCP tool orchestration.

## 2 RELATED WORK

**Agents with Tool Use** ReAct (Yao et al., 2022) integrates reasoning with tool calls, enabling LLM-based agents to interact with external tools. Subsequent research enhanced this capability through fine-tuning (Qin et al., 2023b; Du et al., 2024), modular architectures (Zhuang et al., 2023; Zhou et al., 2024), and retrieval augmentation (Yuan et al., 2024; Zheng et al., 2024). However, these approaches typically rely on ad-hoc integrations where tool schemas must be manually defined and statically injected for each specific model, leading to fragmented ecosystems and limited scalability.

**Model Context Protocol** The Model Context Protocol (MCP) (Anthropic, 2024) addresses these integration challenges by providing a standardized, JSON-RPC-based protocol layer that decouples tool implementation from the agent's core logic. Unlike traditional frameworks that require static tool registration, MCP enables standardized dynamic discovery agents can query and handshake with available tools and resources at runtime from any MCP server without provider-specific configurations. This architecture naturally supports dynamic environments: MCP servers can expose real-time data sources that return time-varying responses, while the standardized interface ensures consistent agent-tool interaction across diverse backends. Since its release, MCP has been rapidly adopted across major AI platforms and has attracted significant research attention (Hou et al., 2025; Ehtesham et al., 2025; Luo et al., 2025; Gao et al., 2025; Liu et al., 2025).

**Evaluation of Agents** Existing tool calling benchmarks predominantly adopted the function-calling paradigm with static tool definitions and mock environments (Yan et al., 2024; Qin et al., 2023b; Guo et al., 2024; Li et al., 2023b; Patil et al., 2024; Wang et al., 2023; Lu et al., 2024; Yao et al., 2024), limiting their ability to evaluate agents in dynamic scenarios where tool outputs vary over time. Consequently, these benchmarks are inadequate for assessing MCP-enabled agents, which must demonstrate the capability to autonomously discover tools and adapt to evolving environments rather than simply strictly following pre-injected schemas.

MCP benchmarks address this limitation, yet significant gaps remain (see Table 1). Early efforts like MCPWorld (Luo et al., 2025), MCP-RADAR (Gao et al., 2025), and MCPEval (Liu et al., 2025) are limited in scope, covering fewer servers or relying on static environments that fail to capture temporal dynamics and cross-domain complexity. While the concurrent LiveMCPBench (Mo et al., 2025) introduces live evaluation, it focuses on simpler tasks (avg. 2.8 steps) and lacks verifiable ground truth, which compromises scoring reliability and may not faithfully reflect relative model capabilities. In contrast, LiveMCP-101 introduces a three-tiered difficulty structure (easy, medium, hard), with tasks requiring an average of 5.4 tool-calling steps, making it a significantly more challenging benchmark. Furthermore, we provide validated execution plans that specify tool names and parameters as explicit references for real-time outputs and trajectory evaluation, improving scoring consistency and showing strong agreement with human raters.

| Benchmark | MCP | Steps | Real Integration | Temporal Dynamics | Verifiable Ground Truth | Cross-Domain |
|---|---|---|---|---|---|---|
| BFCL | - | 3.8 | ✗ | ✗ | ✓ | ✓ |
| $\tau$-Bench | - | - | ✗ | ✗ | ✓ | ✗ |
| ToolSandBox | - | 3.8 | ✗ | ✗ | ✓ | ✓ |
| StableToolBench | - | - | ✗ | ✗ | ✓ | ✓ |
| MCPWorld | 10 | - | ✓ | ✗ | ✓ | ✗ |
| MCP-RADAR | 9 | - | ✗ | ✗ | ✓ | ✗ |
| MCPEval | 19 | - | ✗ | ✓ | ✗ | ✗ |
| LiveMCPBench | 70 | 2.8 | ✓ | ✓ | ✗ | ✓ |
| **LiveMCP-101 (Ours)** | **41** | **5.4** | ✓ | ✓ | ✓ | ✓ |

Table 1: Comparison of LiveMCP-101 with existing benchmarks. Our work distinguishes itself with task complexity (average tool calling steps), real-world integration, and verifiable ground truth.

## 3 LIVEMCP-101

### 3.1 CONSTRUCTION

**Query Generation** To generate high-quality, challenging queries that require agents to leverage multiple MCP servers and tools, we first use GPT-4.1 to sample diverse application domains from the

MCP tool pool spanning 41 servers and 260 tools (server categories in Figure 3a). We then employ the OpenAI o3 model (OpenAI, 2025b) to generate queries of varying complexity, conditioned on domain context and MCP tool specifications (names, descriptions, and parameters). Despite carefully tuned prompts, some generated queries are not solvable with the provided tools or have final outputs that are not easily verifiable. To ensure rigor, we apply multiple rounds of LLM-assisted rewriting with human-in-the-loop revision (Wang et al., 2022), enforcing clarity, balanced difficulty, solvability with the given tools, and objective verifiability. We stratify queries into three difficulty tiers: Easy (30), Medium (30), and Hard (41). Difficulty tiers were determined by author consensus following independent assessments. Assignments were based on two dimensions: (1) reasoning complexity (planning and parameterization), and (2) tool chain length. Easy tasks involve straightforward operations with short tool sequences, while Hard tasks require complex multi-constraint reasoning with extensive orchestration. Figure 2 provides representative examples from each tier.

---

**Easy**

During a recent weekly meeting, my mentor highlighted the need for improved DevOps monitoring. Please prepare a Markdown file named `k8s_issues_report.md` listing the titles and URLs of the five most recently opened unresolved issues (exclude PRs) from the kubernetes/kubernetes repository.

---

**Medium**

As part of a recent initiative at the fictional consultancy firm BrightPath Analytics, commissioned by the renowned artist Lucia Moretti for an upcoming exhibition in Zurich, you are tasked with supporting market research on the digital art landscape. Lucia is specifically interested in public engagement with YouTube content for "AI-generated art tools". Retrieve the first five search results returned for this query. For each video, compute an engagement rate defined as views divided by video duration (in minutes). Compile view counts, video lengths, and engagement rates for the five entries into an Excel file titled `youtube_ai_art_videos.xlsx` for forwarding to Lucias Zurich studio.

---

**Hard**

My 9-year-old son is obsessed with his favorite NBA team and keeps giving me cryptic clues. Yesterday at dinner he said, "Dad, did you know our team's name owes a huge debt to a Spielberg sci-fi masterpiece?" He's been begging me to see a home game at their arena. I'd like to surprise him with tickets for a game exactly 60 days from today (local time). We'll fly in the night before and need accommodation for one night. Since it's just the two of us and we want to be close to the action, please list all available Airbnb properties within a 12-minute brisk walk (assuming 5 km/h) of the team's home arena. My budget is strictly $150–$160 USD per night. Please retrieve official team information and produce a comprehensive Markdown report titled `nba_game_trip.md`. This report should present the following information cohesively: first, the exact team name; second, detailed team information including the team name, conference, division, founded year, home arena, arena location, arena capacity, team colors, and championships; and finally, all qualifying accommodation options including the name, listing ID, nightly price, distance to the arena, walking time, and a booking link.

---

Figure 2: Example queries by difficulty level. These queries require the multi-step composition of heterogeneous MCP tools, with proper parameterization and output handling. Corresponding execution plans are provided in Appendix A.

**Execution Plan Generation** Because tasks interact with live, time-varying MCP services, tool responses are non-stationary over time. Thus, relying on fixed ground-truth answers is unreliable at test time. To address this, we pair each query with an execution plan that specifies a step-by-step sequence of MCP tool invocations with explicit parameterization, and the corresponding Python code to extract the required information from tool-call outputs. We first draft each plan using the o3 model conditioned on the query and tool specifications (names, descriptions, and parameters). We then iteratively revise the plan based on the reference agents execution result and trajectory, combining LLM-assisted edits with targeted manual adjustments to correct logical errors, tool selection, parameterization, and data processing. Approximately 120 expert-hours were required for this revision. Each task is validated across at least three runs with manual verification of correctness. When strictly followed, the finalized plan deterministically yields the correct, time-aligned result relative to the live environment at execution time. The execution plans for the examples in Figure 2 are provided in Appendix A. The distribution of tool-chain lengths across plans is shown in Figure 3b.

## 3.2 EVALUATION

### 3.2.1 EVALUATION FRAMEWORK

For each task, we launch parallel executions: (1) **a real-time reference execution**, where the reference agent strictly follows the validated execution plan to produce the reference output, serving as the dynamic ground-truth answer at evaluation time rather than defining the only correct execution path; and (2) **a real-time test execution**, where the evaluated agent receives the query and a predefined

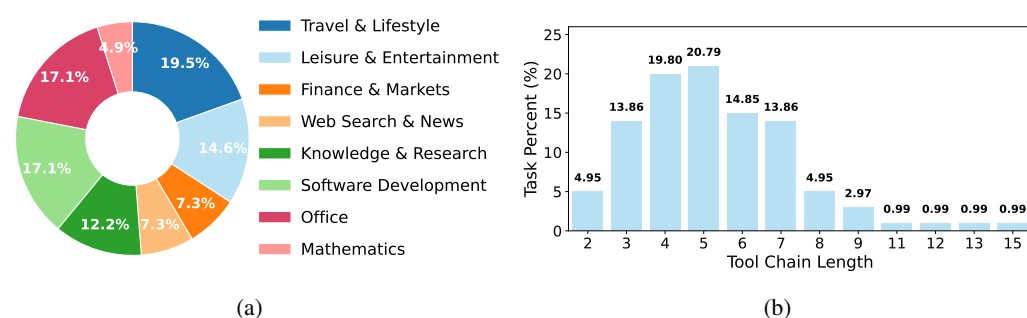

(a)                                                 (b)

Figure 3: Dataset statistics for LiveMCP-101. **(a)** MCP server categories in this work. **(b)** Distribution of tool-chain lengths in the validated execution plans across 101 tasks (mean 5.4 steps, max 15 steps).

per-task MCP pool comprising all task-essential tools plus "distractor" tools (pool construction details in Section 4.1). The evaluated agent independently analyzes the query, selects tools, schedules calls, and processes intermediate results. The test execution runs until the agent declares completion or hits the maximum number of rounds. By executing co-temporally, this setup mitigates temporal drift and enables a fair comparison between the evaluated agents output and the reference. The reference trajectory also supports fine-grained diagnosis of tool-selection, parameterization, and output-handling errors.

### 3.2.2 EVALUATION METRICS

We employ an LLM judge (Zheng et al., 2023) to score the final results and execution trajectories using a 1–5 Likert scale (Liu et al., 2023). Judge prompts are provided in Appendices D.1 and D.2. Let $N$ be the number of tasks. For task $i$, let $s^{\text{res}}(i)$ and $s^{\text{traj}}(i)$ denote the judge-assigned 1–5 scores (Likert) for the final output result and trajectory, and define the normalized scores as $s^{\text{res}}_{\text{norm}}(i) = \frac{s^{\text{res}}(i)-1}{4}$ and $s^{\text{traj}}_{\text{norm}}(i) = \frac{s^{\text{traj}}(i)-1}{4}$, with values in $\{0.00, 0.25, 0.50, 0.75, 1.00\}$. We report the following metrics:

**Task Success Rate (TSR).** TSR is defined as $\frac{1}{N}\sum_{i=1}^{N} \mathbb{1}\{s^{\text{res}}_{\text{norm}}(i) = 1.00\}$, which measures the proportion of tasks that are correctly solved.

**Average Result Score (ARS).** ARS equals $\frac{1}{N}\sum_{i=1}^{N} s^{\text{res}}_{\text{norm}}(i)$, the average normalized result score across tasks, reflecting overall solution quality.

**Average Trajectory Score (ATS).** ATS is computed as $\frac{1}{N}\sum_{i} s^{\text{traj}}_{\text{norm}}(i)$. It assesses execution trajectories for logical coherence, completeness, and correctness, providing a process-level complement to result metrics.

**Average Token Consumption.** For each task, we sum the agent's output tokens across all rounds. The reported value is the mean of these per-task totals over the evaluation set.

**Average Tool Calls.** For each task, we count all tool invocations across the full trajectory. The reported number is the mean of these per-task across the evaluation set.

## 4 EXPERIMENTS

### 4.1 EXPERIMENTAL SETUP

**Models** We evaluate a diverse set of 18 widely used and representative LLMs on LiveMCP-101: OpenAI (GPT-5, GPT-5-mini, GPT-4.1, GPT-4o, GPT-4.1-mini, GPT-4o-mini, o3, o4-mini), Anthropic (Claude-4.1-Opus, Claude-4-Sonnet, Claude-3.7-Sonnet), Google (Gemini-2.5-Pro, Gemini-2.5-Flash), and open-source (Qwen3-235B-A22B, Qwen3-32B, Qwen3-8B, Llama-3.3-70B-Instruct, Llama-3.1-8B-Instruct). For OpenAI reasoning models (OpenAI, 2025b), the reasoning effort is set to medium. For Anthropic models, we evaluate both standard and extended thinking (ET) models (Anthropic, 2025). For Qwen3 models, thinking is enabled by default (Qwen Team, 2025).

**Settings**   Each agent is limited to a maximum of 30 rounds. For reference execution, we employ GPT-4.1 due to its low latency and strong instruction-following capabilities (OpenAI, 2025a), strictly adhering to the validated execution plan to produce the reference output. For each task, a per-task MCP pool is constructed by combining all task-essential servers with randomly sampled MCP servers, yielding a total of 15 MCP servers and 76–125 tools available per task. We adopt the widely used ReAct framework (Yao et al., 2023) for agent execution (prompt in Appendix D.3), and the reference agent prompt is provided in Appendix D.4. GPT-4.1 serves as the LLM judge (Zheng et al., 2023) to evaluate both final results and execution trajectories.

**Metrics**   As described in Section 3.2.2, we report the following metrics for each model: task success rate (TSR), average result score (ARS), average trajectory score (ATS), average output token consumption and average tool calls.

## 4.2   MAIN RESULTS

| Model | Overall | | Easy | | Medium | | Hard | |
|---|---|---|---|---|---|---|---|---|
| | TSR | ARS | TSR | ARS | TSR | ARS | TSR | ARS |
| GPT-5 | **58.42** | **73.02** | **86.67** | **89.17** | **56.67** | **72.50** | **39.02** | **61.59** |
| o3 | 46.53 | 64.60 | 66.67 | 80.00 | 46.67 | 65.83 | 31.71 | 52.44 |
| GPT-5-mini | 43.56 | 63.12 | 63.33 | 82.50 | 43.33 | 64.17 | 29.27 | 48.17 |
| Claude-4.1-Opus (ET) | 41.58 | 61.88 | 56.67 | 79.17 | 43.33 | 61.67 | 29.27 | 49.39 |
| o4-mini | 40.59 | 61.63 | 53.33 | 77.50 | 46.67 | 62.50 | 26.83 | 49.39 |
| Claude-4-Sonnet (ET) | 43.56 | 60.40 | 63.33 | 79.17 | 46.67 | 62.50 | 26.83 | 45.12 |
| Claude-4.1-Opus | 39.60 | 59.41 | 60.00 | 83.33 | 33.33 | 49.17 | 29.27 | 49.39 |
| Claude-4-Sonnet | 37.62 | 55.69 | 63.33 | 78.33 | 46.67 | 65.00 | 12.20 | 32.32 |
| GPT-4.1 | 35.64 | 55.94 | 60.00 | 76.67 | 36.67 | 55.83 | 17.07 | 40.85 |
| Claude-3.7-Sonnet (ET) | 29.70 | 47.77 | 43.33 | 66.67 | 26.67 | 46.67 | 21.95 | 34.76 |
| Gemini-2.5-Pro | 27.72 | 46.78 | 36.67 | 61.67 | 30.00 | 46.67 | 19.51 | 35.98 |
| Claude-3.7-Sonnet | 26.73 | 42.57 | 46.67 | 61.67 | 20.00 | 40.83 | 17.07 | 29.88 |
| Qwen3-235B-A22B | 22.77 | 42.57 | 43.33 | 63.33 | 26.67 | 45.00 | 4.88 | 25.61 |
| GPT-4o | 21.78 | 41.09 | 40.00 | 62.50 | 20.00 | 37.50 | 9.76 | 28.05 |
| GPT-4.1-mini | 17.82 | 35.15 | 36.67 | 56.67 | 13.33 | 31.67 | 7.32 | 21.95 |
| Qwen3-32B | 18.81 | 34.41 | 36.67 | 59.17 | 16.67 | 32.50 | 7.32 | 17.68 |
| GPT-4o-mini | 8.91 | 27.48 | 16.67 | 40.83 | 6.67 | 31.67 | 4.88 | 14.63 |
| Gemini-2.5-Flash | 10.89 | 22.48 | 26.67 | 44.17 | 10.00 | 22.33 | 0.00 | 6.71 |
| Qwen3-8B | 3.96 | 11.63 | 10.00 | 26.67 | 3.33 | 8.33 | 0.00 | 3.05 |
| Llama-3.3-70B-Instruct | 1.98 | 6.93 | 3.33 | 15.83 | 3.33 | 5.83 | 0.00 | 1.22 |
| Llama-3.1-8B-Instruct | 0.99 | 2.72 | 3.33 | 9.17 | 0.00 | 0.00 | 0.00 | 0.00 |

Table 2: Task success rate (TSR, %) and average result score (ARS, %) reported overall and by difficulty (Easy/Medium/Hard). Shaded rows mark the top-3 models by overall TSR. Bold indicates best in each column. ET denotes extended thinking enabled for Anthropic models.

As shown in Table 2, GPT-5 achieves the best overall performance on LiveMCP-101, leading across all difficulty tiers. Ranking next are o3, GPT-5-mini, Claude-4.1-Opus (ET), and Claude-4-Sonnet (ET), indicating that stronger reasoning effort can yield meaningful improvements for dynamic, multi-step problem solving and MCP tool use. Among mid-tier proprietary models, GPT-4.1, Gemini-2.5-Pro, and Claude-3.7-Sonnet perform reasonably well but trail the top performers. Open-source models lag behind closed-source models. Among open-source models, Qwen3-235B-A22B achieves the best performance (TSR/ARS: 22.77%/42.57%), yet remains far from the frontier. Llama models underperform on LiveMCP-101, and a detailed analysis is provided in Section 5.2. Performance degrades substantially with task difficulty across all models. Notably, even the strongest model attains only 39.02% TSR on Hard. Rankings by TSR and ARS are broadly consistent.

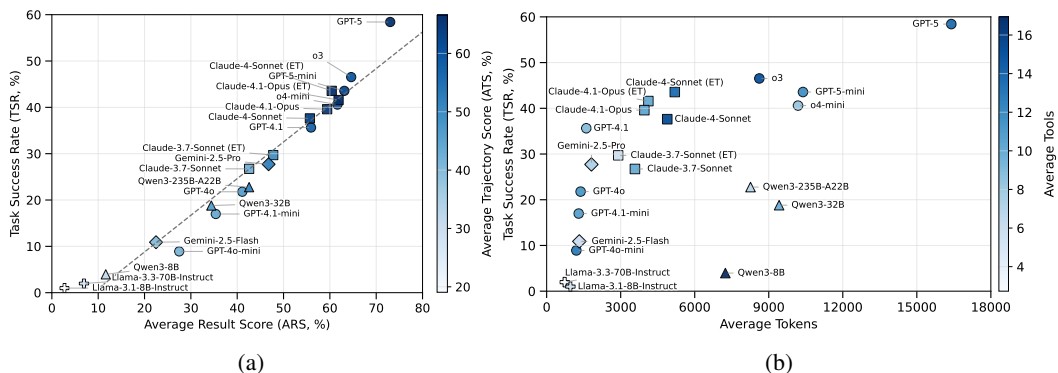

(a)             (b)

Figure 4: Results on LiveMCP-101, showing model performance in terms of task success rate (TSR), average result score (ARS), average trajectory score (ATS), average token consumption, and average tool calls. **(a)** TSR (%) vs. ARS (%), with color encoding ATS (%). **(b)** TSR (%) vs. average tokens per task, with color encoding average tool calls. In both plots, marker shapes denote model families.

Figure 4a visualizes the relationship among TSR, ARS, and ATS. The color-encoded ATS increases with both ARS and TSR, with higher-ATS models clustering toward the upper-right region. This indicates an overall positive association between trajectory quality and output quality. Higher ATS corresponds to more reliable tool selection, parameterization, and post-processing, which thereby helps satisfy the task success criteria. Figure 4b shows the relationship between TSR, the average number of output tokens, and the average number of tool calls per task. Closed-source models exhibit a mild upward trend with tokens, yet planning quality remains the primary driver. In contrast, open-source models exhibit two characteristic inefficiencies. Llama variants cluster in the low-token, low-tool region, under-exploring tool affordances and often stopping early, which yields low ARS and TSR. Qwen variants trend toward the opposite extreme, producing longer outputs and invoking more tools without commensurate gains compared to the closed-source models. Extended-thinking variants consistently shift the efficiency frontier upward at comparable token budgets, suggesting gains from improved planning and error recovery rather than verbosity.

### 4.3 ABLATION STUDY

We conduct ablations on GPT-5, Claude-4.1-Opus (ET), GPT-4.1, Gemini-2.5-Pro, Qwen3-235B-A22B, and Qwen3-8B, covering frontier, mid-tier closed-source models and open-source models.

**Impact of maximum iteration rounds** In LiveMCP-101, the longest validated execution plan requires 15 tool calls. By default, each agent is limited to 30 iteration rounds, where each round may involve one or more tool invocations. To study sensitivity to the iteration budget, we vary the maximum number of rounds to 15, 20, 25, and 50. All other settings are held fixed as in Section 4.1. The results in Figure 5 (panels a-b) highlight two key phenomena. First, increasing the max iteration limit from 15 up to 25 rounds consistently improves task success rate, as the added budget enables more thorough tool exploration and error recovery (Yuan et al., 2025; Zhang et al., 2025b). Notably, although the longest validated execution plan comprises 15 tool calls (with an average of 5.4), the continued gains when raising the round limit from 15 to around 25 indicate that agents often expend additional rounds on error recovery or redundant deliberation, even on correctly solved instances, revealing headroom to improve execution efficiency. Second, beyond 25 rounds, gains saturate: performance becomes largely constrained by model capability, particularly planning quality and tool-use competence, rather than round capacity. Additional rounds yield diminishing returns and may introduce noise or compound errors, thereby leaving performance essentially flat.

**Impact of the number of MCP servers** In the default setting, the most demanding task requires up to 6 MCP servers, and we expose a per-task MCP pool of 15 servers to the evaluated agent. To study sensitivity to MCP server breadth, we vary the pool size to 6, 10, 12, and 15. We set 15 as the upper limit because larger pools could hit tool number limits (e.g., 128 tools per request) (OpenAI, 2025) or exceed context length. This choice keeps the setup realistic and comparable to real-world

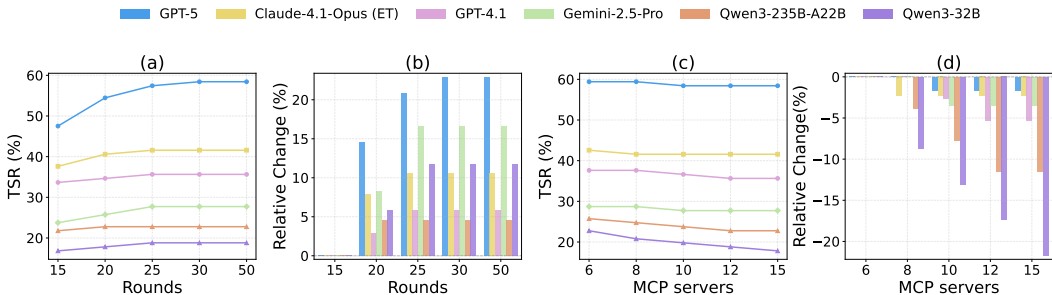

Figure 5: Ablation study results. **(a)** TSR (%) vs. maximum iteration rounds: all models improve from 15 to approximately 25 rounds, then plateau. **(b)** Relative TSR change (%) with respect to the 15-round setting shows diminishing returns beyond about 25. **(c)** TSR (%) vs. number of MCP servers in the per-task pool: top-tier models remain largely stable, while weaker or mid-tier models degrade as distractors grow. **(d)** Relative TSR change (%) with respect to the 6-server setting shows that larger pools affect weaker models more, consistent with long-context sensitivity and tool-selection noise.

deployments. As the pool grows, the expanded tool search and tool call space increases selection overhead and the likelihood of spurious tool usage. Weaker and mid-tier models are more sensitive to this effect, often showing declines as distractors accumulate and planning bandwidth is diluted. In contrast, top-tier systems (e.g., GPT-5, Claude-4.1-Opus (ET)) remain largely stable: stronger planning and tool-screening mitigate distractors, so performance changes are negligible.

### 4.4 ANALYSIS OF LLM-AS-A-JUDGE

We apply an LLM-as-a-Judge to score both final results and execution trajectories. To assess reliability, we conduct a blinded human-expert study on a stratified subset of tasks for six representative models: GPT-5, Claude-4.1-Opus (ET), GPT-4.1, Gemini-2.5-Pro, Qwen3-235B-A22B, and Qwen3-32B. Experts follow the same rubric and judge prompts as the LLM judge. We compare human and LLM-judge decisions at the per-task level and report inter-rater agreement using quadratic-weighted Cohen's $\kappa$ (Cohen, 1960). We evaluate a sampled set of 30 tasks in total with 10 per difficulty

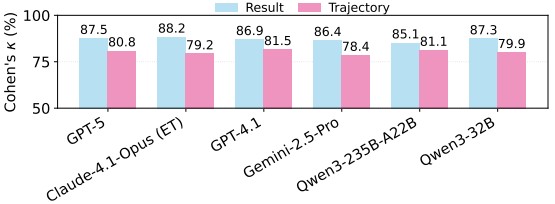

Figure 6: Human–LLM agreement (Cohen's $\kappa$, %) on result and trajectory evaluation for six models. Blue bars denote scores for the result evaluation, and pink bars denote scores for the trajectory evaluation.

tier (Easy, Medium, Hard). Across all six models, the human vs. LLM-judge agreement (quadratic-weighted Cohen's $\kappa$) exceeds 0.85 for the result evaluations and 0.78 for the trajectory evaluations respectively, indicating consistent, human-aligned ratings (Landis & Koch, 1977; Fleiss, 1981).

To further validate judge stability, we evaluated the same six models using three different LLM judges (GPT-4.1, Claude-4-Sonnet, Gemini-2.5-Pro) across all 101 tasks. The results in Table 3 show high correlation across judges, confirming the robustness of our evaluation framework. We attribute this high consistency to the nature of our tasks, which rely on objective, verifiable answers. Consequently, the primary role of the LLM judge is limited to handling formatting differences in the responses, minimizing subjective variance.

## 5 DISCUSSION

### 5.1 TOKEN EFFICIENCY

We observe that closed-source models exhibit a log-shaped trend: task success rate (TSR) rises rapidly with small token budgets, then plateaus (Figure 4b). Intuitively, early tokens drive high-value actions, such as planning, probing tools, checking constraints, yielding large gains. As budgets grow, additional tokens primarily add redundancy (longer explanations, repeated self-checks) rather than new evidence, yielding diminishing marginal returns. This trend reflects token efficiency: even top-performing models show diminishing improvements beyond moderate token budgets. Maximizing

| Model | GPT-4.1 | | Claude-4-Sonnet | | Gemini-2.5-Pro | | Std. Dev. | |
|---|---|---|---|---|---|---|---|---|
| | TSR | ARS | TSR | ARS | TSR | ARS | TSR | ARS |
| GPT-5 | **58.42** | **73.02** | **57.43** | **73.68** | **58.42** | **74.57** | **0.57** | **0.78** |
| Claude-4.1-Opus | 41.58 | 61.88 | 40.59 | 61.45 | 41.58 | 63.08 | 0.57 | 0.87 |
| GPT-4.1 | 35.64 | 55.94 | 35.64 | 55.07 | 37.62 | 56.36 | 1.14 | 0.66 |
| Gemini-2.5-Pro | 27.72 | 46.78 | 27.72 | 45.12 | 28.71 | 47.25 | 0.57 | 1.10 |
| Qwen3-235B | 22.77 | 42.57 | 20.79 | 42.41 | 22.77 | 42.76 | 1.14 | 0.19 |
| Qwen3-32B | 18.81 | 34.41 | 18.81 | 34.91 | 19.80 | 33.62 | 0.57 | 0.66 |

Table 3: Cross-validation of LLM judges: Task success rate (TSR, %) and average result score (ARS, %) evaluated by three different LLM judges (GPT-4.1, Claude-4-Sonnet, and Gemini-2.5-Pro). The low Standard Deviation (SD) across judges demonstrates the robustness of our evaluation framework.

intelligence per token remains an important open challenge for MCP-enabled agents. In contrast to closed-source models, open-source models deviate from this trend: at comparable or higher token budgets, their TSR is substantially lower, suggesting difficulty in converting tokens into reliable evidence and thus lower token efficiency.

## 5.2 FAILURE ANALYSIS

To diagnose failure modes in MCP-based tool use, we analyze execution logs across different models and identify three error categories with seven subtypes: **tool planning and orchestration errors** (1–4), **parameter errors** (5–6), and **output handling errors** (7).

(1) **Ignoring requirement**: the agent misses an explicitly stated requirement and does not select any relevant tool. Typical signs include no corresponding thinking process and tool call, early termination, or a generic final answer that does not address the requirement. This often occurs when the agent fails to extract key requirements from the prompt or loses track of them during execution. (2) **Overconfident self-solving**: the agent recognizes the requirement but attempts to answer from its own knowledge or using its own reasoning and capabilities without calling the needed tool. Indicators include the absence of a corresponding tool call, generic or hallucinated answers, and premature termination. (3) **Unproductive thinking**: the agent acknowledges the need for a tool and may propose plans or parameters, but executes no relevant invocations and produces no step that satisfies the requirement. The trajectory loops in verbose planning and terminates (either prematurely or at the round limit) without any relevant invocation. Typical signs are repeated plan rewrites without execution and excessively token-consuming thinking. (4) **Wrong tool selection**: the agent does invoke a tool but selects an inappropriate one, leading to erroneous intermediate states or final outputs. Errors may occur as a one-off mistake or as repeated incorrect selections that exhaust the iteration budget. Indicators include irrelevant responses, repeated mistakes, or missing required fields in outputs. (5) **Syntactic errors**: parameters provided to a tool are malformed, such as having incorrect types, missing or wrong field names, or invalid schema. These errors prevent the MCP server from correctly parsing the request, leading to failure. (6) **Semantic errors**: parameters are well-formed but misaligned with task intent. Common cases include inappropriately scoped query strings, incorrect identifiers or entity references, and wrong contextual constraints. These errors often arise from mistakes in intermediate reasoning used to generate parameters. (7) **Output parsing errors**: the tool returns a correct result, but the agent fails to parse or transform it correctly, yielding incorrect intermediate states or final answers. We further evaluate representative models spanning a range of capabilities. Figure 7 summarizes the distribution of error types, and Appendix B provides illustrative examples for each type. We highlight the following findings:

- Semantic errors are the dominant failure mode. Top-performing models exhibit rates of 16–25%, while lower-performing models exceed 40% (e.g., GPT-4.1-mini), indicating that content grounding and constraint enforcement are primary bottlenecks in real-world tool use.

- Syntactic errors are negligible for frontier models but reach approximately 48% for Llama-3.3-70B-Instruct. A plausible cause is limited MCP-specific training: MCP adoption surged (Ehtesham et al., 2025) after the Llama-3 release (Meta Llama Team, 2024), suggesting that targeted fine-tuning on MCP tool-call schemas could help reduce such errors and improve overall performance.

- Overconfident self-solving is more common in mid-tier models. They are capable enough to attempt self-solving, but not always discerning enough to know when a tool is necessary. Under the cognitive load of large tool pools and long contexts, planning and screening remain brittle, making reliance on internal knowledge (Chhikara, 2025) seem safer than attempting uncertain tool selection and parameterization.

- Output parsing errors are a persistent and pervasive "last-mile" problem. This typically occurs when models struggle to handle structured data, failing to accurately extract key information from complex formats like JSON. Such failures are particularly revealing, as they underscore the critical difference between merely invoking a tool and the skill of interpreting its results.

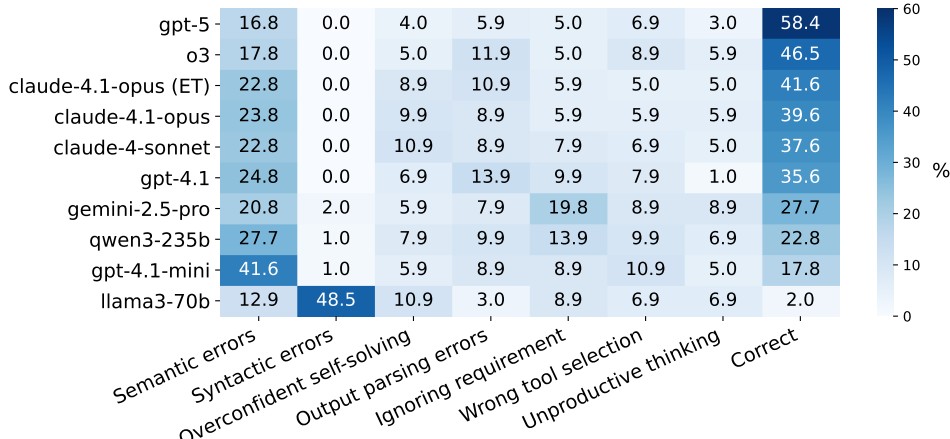

Figure 7: Error classification across models. The rightmost column "Correct" indicates TSR, while the remaining columns decompose failures into seven subtypes. Each row represents a model, and each cell shows the percentages over all instances, darker color indicates higher percentages.

# 6 CONCLUSION

In this work, we introduce **LiveMCP-101**, a benchmark of 101 real-world, multi-step tasks evaluating agents MCP-based tool use in dynamic environments. We propose a parallel evaluation protocol anchored in validated execution plans that mitigates temporal drift and enables robust, comparable scoring of both final outputs and execution trajectories. Experiments across 18 models show that even frontier LLMs attain a task success rate below 60%, revealing substantial gaps in tool planning, parameterization, and output handling. A fine-grained error analysis reveals seven failure modes, guiding targeted advances toward more reliable and capable agents for multi-step tool orchestration.

## 7 ETHICS STATEMENT

The authors of this paper have read and agree to abide by the ICLR Code of Ethics. We do not foresee any direct negative societal impact from our work.

## 8 REPRODUCIBILITY STATEMENT

We are committed to ensuring the reproducibility of our research. To support this, we provide implementation details in our paper. We commit to making our source code publicly available upon acceptance of the paper.

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

## A   EXAMPLE EXECUTION PLANS

---

**Execution Plan**

Step 1: **Tool**: `github.search_issues`
**Params**:   `{query = "repo:kubernetes/kubernetes is:issue is:open", sort = "created", order = "desc", per_page = 5, page = 1}`
**Purpose**: Fetch the list of open issues from the kubernetes/kubernetes repository.

Step 2: **Tool**: `filesystem.write_file`
**Params**:   `{path = "k8s_issues_report.md", content = "markdown-formatted list with titles and links"}`
**Purpose**: Write the list of open issues to a markdown file.

---

Figure 8: Execution plan for the "Easy" example in Figure 2. Each step specifies the MCP tool, concrete parameters, and its purpose. During evaluation, the reference agent strictly follows the plan, which deterministically produces the real-time reference output used for scoring.

---

**Execution Plan**

Step 1: **Tool**: `youtube-data.searchVideos`
**Params**: `{query = "AI art generator tutorial", maxResults = 5}`
**Purpose**: Search for videos related to AI art tools.

Step 2: **Tool**: `youtube-data.getVideoDetails`
**Params**: `{videoIds = [<top_5_video_ids>]}`
**Purpose**: Extract detailed info for the top 5 videos.

Step 3: **Tool**: `code-interpreter.execute_python_code`
**Params**:   `{code = "Python script to compute engagement rates (views per minute) for top 5 videos and output JSON sorted by engagement_rate."}` *(full code omitted for brevity)*
**Purpose**: Analyze video data to compute engagement rates.

Step 4: **Tool**: `excel.create_workbook`
**Params**: `{filepath = "youtube_ai_art_videos.xlsx"}`
**Purpose**: Create the Excel workbook.

Step 5: **Tool**: `excel.write_data_to_excel`
**Params**:   `{filepath = "youtube_ai_art_videos.xlsx", sheet_name = "AI Art Analysis", data = [[ "Video Title", "Channel", "Views", "Duration (min)", "Engagement Rate (Views/Min)", "URL" ]] + [[ v['title'], v['channel'], v['views'], v['duration_minutes'], v['engagement_rate'], v['url'] ] for v in <processed_videos_from_step_3>], start_cell = "A1"}`
**Purpose**: Write titles, channels, views, durations, engagement rates, and URLs to Excel.

---

Figure 9: Execution plan for the "Medium" example in Figure 2. Each step specifies the MCP tool, concrete parameters, and its purpose. During evaluation, the reference agent strictly follows the plan, which deterministically produces the real-time reference output used for scoring.

**Execution Plan**

Step 1: **Tool**: `time.get_current_time`

**Params**: {timezone = "America/Toronto"}

**Purpose**: Get current date (ET) to derive check-in (today+59d) and check-out (today+60d).

Step 2: **Tool**: `code-interpreter.execute_python_code`

**Params**: {code = "Python to parse [current_time], compute checkin=today+59d, checkout=today+60d, output JSON as {checkin, checkout, current_date}"}

**Purpose**: Calculate dates for the Airbnb stay and return `dates`.

Step 3: **Tool**: `airbnb.search`

**Params**: {location = "downtown Toronto, Toronto, Ontario, Canada", checkin = [dates.checkin], checkout = [dates.checkout], adults = 1, children = 1, infants = 0, pets = 0, minPrice = 150, maxPrice = 160, cursor = null}

**Purpose**: Search Airbnb properties near Scotiabank Arena for 2 guests within budget.

Step 4: **Tool**: `google-maps.maps_distance_matrix`

**Params**: {origins = coordinates extracted from airbnb_results, destinations = [[43.6435, -79.3791]], mode = "walking"}

**Purpose**: Compute walking distance/time from each listing to Scotiabank Arena.

Step 5: **Tool**: `code-interpreter.execute_python_code`

**Params**: {code = "Python to load [airbnb_results], [distances_to_arena], [raptors_team_info], [dates]; filter listings within 1km (12-min walk at 5km/h), sort by distance, and compose Markdown with team info, trip dates, and qualifying properties; print JSON {markdown_content}"} *(full code omitted for brevity)*

**Purpose**: Process data and generate the final Markdown report content.

Step 6: **Tool**: `filesystem.write_file`

**Params**: {path = "nba_game_trip.md", content = [report_content.markdown_content]}

**Purpose**: Write the comprehensive Markdown report to file.

Figure 10: Execution plan for the "Hard" example in Figure 2. Each step specifies the MCP tool, concrete parameters, and its purpose. During evaluation, the reference agent strictly follows the plan, which deterministically produces the real-time reference output used for scoring.

## B  FAILURE CASES

This section presents representative failure cases observed in our evaluation. Each case is organized into three parts: (1) **Task Query**, (2) **Agent Trajectory**, and (3) **Error Diagnosis**. The **Agent Trajectory** includes both the model's *Think* steps and the corresponding *Tool Call* invocations. For readability, the *Think* content is lightly abridged to retain only the core reasoning.

### B.1  IGNORING REQUIREMENT

---

**Task Query (Hard)**

During the monthly technology review session at the San Francisco headquarters of enterprise AI consultancy Pacific Intelligence Partners, Lead AI Architect Dr. Sarah Chen was assigned to develop a comprehensive market intelligence report on cutting-edge language model orchestration frameworks. As part of their strategic evaluation for a Fortune 500 technology client exploring conversational AI implementations, Dr. Chen has been specifically directed to analyze the open-source language model application framework known as "LangChain", which has gained remarkable traction in the enterprise AI development community for its sophisticated chain-of-thought capabilities. Dr. Chen requires an extensive analysis that identifies the primary developer driving recent project momentum by examining commit patterns within the latest 100 contributions to understand development leadership dynamics. The investigation should extend to profiling this key contributor's broader open-source involvement, specifically highlighting their most and second most starred public repository to assess their technical expertise and community influence, including both project name and repository link for further evaluation. Additionally, the research must establish basic developer identification information for this key contributor including their unique GitHub identifiers. All findings must be synthesized into a structured JSON document titled `langchain_competitive_analysis.json` that will serve as the foundation for Pacific Intelligence Partners' AI framework adoption recommendations, with Dr. Chen emphasizing that precision and comprehensiveness are critical given the strategic importance of this assessment.

---

**Agent Trajectory**

**Think:** First, I need to locate the LangChain GitHub repository.
**Tool Call:**

```
{
  "name": "github.search_repositories",
  "parameters": { "query": "langchain", "max_results": 10 }
}
```

**Tool Call Result:** Found repository `langchain-ai/langchain`.

**Think:** Fetch the latest 100 commits to infer the most active recent contributor.
**Tool Call:**

```
{
  "name": "github.list_commits",
  "parameters": { "owner": "langchain-ai", "repo": "langchain", "page": 1, "per_page": 100
        }
}
```

**Tool Call Result:** Retrieved recent commits; a top contributor username is inferred.

**Think:** Get the contributor's top repositories by stars.
**Tool Call:**

```
{
  "name": "github.search_repositories",
  "parameters": { "query": "user:<top_contributor_username> sort:stars-desc", "max_results":
        2 }
}
```

**Tool Call Result:** Two repositories returned (most- and second-most-starred).

**Think:** Write the competitive analysis JSON file.
**Tool Call:**

---

```
{
  "name": "filesystem.write_file",
  "parameters": {
    "path": "langchain_competitive_analysis.json",
    "content": "{ \"github_id\": \"<username>\", \"most_starred_repo\": \"<repo_1>\", \"
        most_starred_link\": \"<url_1>\", \"second_most_starred_repo\": \"<repo_2>\", \"
        second_most_starred_link\": \"<url_2>\" }"
  }
}
```
❌ Error

**Tool Call Result:** File successfully saved.

## Error Diagnosis

**Ignoring requirement:** The agent stopped after minimal GitHub queries and returned a non-conforming JSON, missing project analysis and the lead contributor's identifiers, etc.

### B.2 OVERCONFIDENT SELF-SOLVING

## Task Query (Medium)

Dr. Aisha Patel, lead researcher at the Institute for Advanced Computational Science, is preparing the final benchmark for her white-paper on stiff, highly oscillatory transcendental equations. She has chosen the function $f(x) = e^{0.2x} \cdot \sin x + J_0(x) - x^4 + 2x^2 - 1$, where $J_0$ is the Bessel function of the first kind. Your assignment is to locate every real root of $f(x)$ on the closed interval $[-2, 2]$ accurate to at least $1 \times 10^{-10}$. For each root you must determine: (1) whether it is *simple* or *multiple*; and (2) the local curvature sign: evaluate $f''(x)$; label the root concave_up if $f''(x) > 0$, concave_down if $f''(x) < 0$, or flat if $|f''(x)| \leq 1 \times 10^{-6}$. Finally compute $S = \sum 1/(1 + x_i^2)$ over all roots $x_i$. Produce a JSON file named `bessel_equation_analysis.json` with exactly the structure:

```
{
  "roots": [
    {"x": <root_1>, "multiplicity": "simple|multiple", "curvature": "concave_up|concave_down
        |flat"},
    {"x": <root_2>, "multiplicity": "...", "curvature": "..."}
  ],
  "S": <sum_value>
}
```

No additional text may appear in the file.

## Agent Trajectory

**Think:**

```
{
  "roots": [
    { "x": -1.4759783916, "multiplicity": "simple", "curvature": "concave_up" },
    { "x": -0.7731836810, "multiplicity": "simple", "curvature": "concave_down" },
    { "x": 0.7731836810, "multiplicity": "simple", "curvature": "concave_up" },
    { "x": 1.4759783916, "multiplicity": "simple", "curvature": "concave_down" }
  ],
  "S": 1.0389754276
}
```
❌ Error

**Tool Calls:** None (the model directly outputs final results without calling any tools).

## Error Diagnosis

**Overconfident self-solving:** The agent was overconfident and failed to invoke the MCP tool `wolfram-alpha` to solve the equation; instead, it directly produced roots, resulting in incorrect final results.

## B.3 UNPRODUCTIVE THINKING

**Task Query (Medium)**

As part of an ongoing financial literacy initiative at the Tokyo branch of the multinational consulting firm Nakamura & Klein, Senior Analyst Mei Tanaka has been tasked with preparing a comprehensive purchasing power parity (PPP) valuation analysis for inclusion in the firm's upcoming internal newsletter. During a recent virtual meeting with her colleague Lukas Schneider in Frankfurt, Mei agreed to focus specifically on the exchange rate relationships between the Japanese Yen (JPY), the Euro (EUR), and the US Dollar (USD) using the "Big Mac Index" as a simplified model, given its accessibility for non-specialist audiences. For this analysis, use the following Big Mac prices in local currencies: USA: \$5.69, Japan: 790, Germany: 4.95. Calculate the *implied PPP exchange rate* for each currency and determine the degree to which JPY and EUR are overvalued or undervalued relative to USD, expressed as a percentage. Finally, produce a concise analysis in the form of a JSON file titled `ppp_analysis.json` containing the currency comparison table with Market Rate, Implied PPP Rate, and Valuation percentage for each currency.

**Agent Trajectory**

**Think:** Okay, let's tackle this step by step. We need a PPP analysis (Big Mac Index) for JPY/EUR vs USD and output a JSON file. First compute implied PPP rates from prices: $\text{PPP}_{\text{JPY/USD}} = 790/5.69$, $\text{PPP}_{\text{EUR/USD}} = 4.95/5.69$. Then compare to market rates to get valuation $\big((\text{Market} - \text{PPP})/\text{PPP}\big) \times 100$. ... We must fetch current market rates via `exchange-rate.get_exchange_rate` (USD→JPY, USD→EUR), ensure quote direction matches PPP (local per USD), and then compute in `code-interpreter` for accuracy. ... Clarify sign: if Market > PPP for JPY/USD, JPY is *undervalued*; if Market < PPP, *overvalued*. Similar logic for EUR/USD with EUR per USD. ... After computing, assemble JSON with `analysis_date`, `methodology`, and an array of {`currency`, `market_rate`, `implied_ppp_rate`, `valuation_percent`}. Save to `ppp_analysis.json`.

**Tool Calls (required but missing):**

```
                                                                    ❌ Error
{
  "tool_calls": []
}
```

**Tool Call Result:** No market rates retrieved, no PPP/valuation computed, and no JSON written.

**Error Diagnosis**

**Unproductive thinking:** The agent enumerated a correct plan but failed to execute the necessary tools (e.g., `exchange-rate.get_exchange_rate`, `code-interpreter.execute_python_code`, `filesystem.write_file`), resulting in no market rates, no PPP/valuation computation, and no deliverable.

## B.4 WRONG TOOL SELECTION

**Task Query (Easy)**

As part of my ongoing efforts to improve the wellbeing initiatives for our marketing team, I have been collaborating with Ms. Du, our HR coordinator, who recently suggested that we incorporate outdoor activities for remote employees based in major metropolitan areas. In preparation for a proposed wellness pilot program, I am evaluating viable walking routes for our colleagues, such as Michael Chen, who is currently based in the city in the Eastern part of the States whose Asian population exceeded 1 million in the year before the world population surpassed 7 billion, and has expressed interest in participating tomorrow. To ensure the success of this initiative, could you please retrieve the weather forecast for this city for tomorrow and create a Markdown file titled `city_walk_recommendation.md` that includes tomorrow's date (in the city's timezone), a weather summary at 10:00 AM (low and high temperature, wind speed, and a walking recommendation clearly stating whether it's good or bad weather for walking). This deliverable will be included in my report to the management committee.

**Agent Trajectory**

**Think:** Infer the city as New York City (Eastern U.S., Asian population > 1M in 2010). Obtain tomorrow's date in the city's timezone.
**Tool Call:**

```
{
  "name": "time.get_current_time",
  "parameters": { "timezone": "America/New_York" }
}
```

**Tool Call Result:** Retrieved today's date and computed tomorrow's date (city timezone).
**Think:** Retrieve the forecast for New York City at 10:00 AM tomorrow (needs day selection and time resolution).
**Tool Call:**

```
{
  "name": "weather.get_weather",
  "parameters": { "city": "New York" }
}
```
❌ Error

**Tool Call Result:** Returned a generalized report without a tomorrow filter or 10:00 AM granularity.

**Think:** Write the Markdown recommendation to the required directory.
**Tool Call:**

```
{
  "name": "filesystem.write_file",
  "parameters": {
    "path": "city_walk_recommendation.md",
    "content": "City Walk Recommendation"
  }
}
```

**Tool Call Result:** File written successfully.

### Error Diagnosis

**Wrong tool selection:** The agent invoked `weather.get_weather` (current/general conditions) instead of `weather_get_forecast` with explicit day/time selection (tomorrow 10:00, timezone-aligned). This yielded non-target temporal data and a misaligned report.

## B.5 SYNTACTIC ERRORS

### Task Query (Easy)

In preparation for the upcoming quarterly meeting at the headquarters of Aurora Cinematics, where CEO Dr. Eleanor Matthews and her executive team—including Chief Content Officer Rajiv Patel and Marketing Director Sofia Nguyen—will review recent entertainment trends, you have been assigned to assist the analytics division based in the London office. As part of the preparatory materials for the boardroom discussion, please create a markdown report titled `trending_movies_report.md` that enumerates the top 5 trending movies for the current week. For each movie entry, include a synopsis, a list of the main cast members, and a direct link to the official trailer on YouTube (search "<Movie Title> Official trailer" and use the first result). This report will be referenced during the strategic planning session scheduled in Conference Room B, adjacent to the office's iconic glass installation commemorating the company's founding.

### Agent Trajectory

**Think:** To create a markdown report titled `trending_movies_report.md` that enumerates the top 5 trending movies for the current week, including each movies synopsis, main cast, and a direct YouTube trailer link, we will follow these steps.
1. Search for the top 5 trending movies of the week.
2. Get the details of each movie, including synopsis and main cast.
3. Find the YouTube trailer for each movie.
4. Create a markdown file with the required information.
Let's start by searching for the top 5 trending movies of the week.
**Tool Call:**

```
{
    "name": "tmdb-movies.get_popular_movies",
    "parameters": {
        "page": "1"
    }
}
```
❌ Error

**Tool Call Result:** Input validation error: '1' is not of type 'integer'.

**Think:** After getting the list of popular movies, we'll get the details of each movie. We will assume that the first 5 movies in the list are the top 5 trending movies for this week.
**Tool Call:**

```
{
    "name": "tmdb-movies.get_movie_details",
    "parameters": {
        "movie_id": "ID of the first movie"
    }
}
```

(Repeat the above step for the remaining 4 movies.)
**Tool Call Result:** Input validation error: 'ID of the first movie' is not of type 'integer'.

**Think:** Next, we need to find the YouTube trailer for each movie. We can use the movie title to search for the trailer on YouTube.
**Tool Call:**

```
{
    "name": "youtube-data.searchVideos",
    "parameters": {
        "query": "Title of the first movie trailer",
        "maxResults": "1"
    }
}
```

(Repeat the above step for the remaining 4 movies.)
**Tool Call Result:** Failed to call tool 'searchVideos' on server 'youtube-data': Invalid arguments for tool searchVideos.

**Think:** Finally, we create a markdown file with the required information.
**Tool Call:**

```
{
    "name": "filesystem.write_file",
    "parameters": {
        "path": "trending_movies_report.md",
        "content": "Markdown content with movie details and trailer links"
    }
}
```

**Tool Call Result:** Successfully wrote to trending_movies_report.md.

---

### Error Diagnosis

**Syntactic errors:** The initial call to `tmdb-movies.get_popular_movies` passed `page` = '1' (string) instead of the required integer, triggering the message Input validation error: '1' is not of type 'integer', and preventing retrieval of valid movie IDs. Consequently, subsequent steps propagated placeholders and wrong types (e.g., `tmdb-movies_get_movie_details` with 'ID of the first movie'), causing cascading failures.

## B.6    SEMANTIC ERRORS

### Task Query (Hard)

My wife just accepted a position at the arena of an NBA franchise whose team name does not include either its home city or state. After her shifts, she likes to swing by one of three favourite shops but only gave me riddles to identify them: the first is a chain that lends its name to an NBA arena, the second belongs to a retail group that (as of 2025) is even larger than the first chain's parent company, and the third is a membership-based warehouse club founded in the same city where the world's richest person created his company back in 2006. Could you figure out which three retailers those clues point to, then locate the single nearest branch of each one to her workplace and list, for every store, its name, street address, and opening hours in a markdown file named `nearest_stores.md`?

### Agent Trajectory

**Think:** First, identify the NBA team (and arena) whose team name does not include its home city or state. Assume Golden State Warriors and use Chase Center as the workplace anchor (unverified). Search the arena address.
**Tool Call:**

```
{
  "name": "google-maps.maps_search_places",
  "parameters": { "query": "Chase Center arena San Francisco" }
}
```

**Tool Call Result:** Found "Chase Center, 1 Warriors Way, San Francisco, CA 94158, USA" as anchor.

**Think:** Solve the retailer riddles. Retailers identified as: Chase (Chase Bank); Walmart; Costco (membership warehouse club founded in Seattle).
**Think:** From the assumed anchor, search the nearest branches for each retailer.
**Tool Call:**

❌ Error

```
{
  "name": "google-maps.maps_search_places",
  "parameters": {
   "query": "Chase Bank",
   "location": { "latitude": 37.7680183, "longitude": -122.3878772 },
   "radius": 5000
  }
}
```

**Tool Call:**

```
{
  "name": "google-maps.maps_search_places",
  "parameters": {
   "query": "Walmart",
   "location": { "latitude": 37.7680183, "longitude": -122.3878772 },
   "radius": 20000
  }
}
```

**Tool Call:**

```
{
  "name": "google-maps.maps_search_places",
  "parameters": {
   "query": "Costco",
   "location": { "latitude": 37.7680183, "longitude": -122.3878772 },
   "radius": 20000
  }
}
```

**Tool Call Result:** Retrieved nearby candidates for Chase Bank, Walmart, and Costco (relative to the assumed anchor).

**Think:** Fetch place details (name, address, opening hours) for the top candidates.
**Tool Call:**

```
{
  "name": "google-maps.maps_place_details",
  "parameters": { "place_id": "ChIJw84d8S5_j4ARCcEUT8Hiw78" }
}
```

**Tool Call:**

```
{
  "name": "google-maps.maps_place_details",
  "parameters": { "place_id": "ChIJQZSXnWGFj4ARDAoMDFAsCAE" }
}
```

**Tool Call:**

```
{
  "name": "google-maps.maps_place_details",
  "parameters": { "place_id": "ChIJC2juwih-j4ARvLq4f6Oyeuc" }
}
```

**Tool Call Result:** Fetched details (names, addresses, hours) for the three retailers (relative to the assumed anchor).

**Think:** Write the results to a markdown file.
**Tool Call:**

```
{
  "name": "filesystem.write_file",
  "parameters": {
   "path": "nearest_stores.md",
   "content": "Nearest Stores to Chase Center"
  }
}
```

**Tool Call Result:** Successfully wrote to `nearest_stores.md`.

---

**Error Diagnosis**

**Semantic errors:** The arena-named chain should be *Target* (Target Center), but the agent resolved it as *Chase*, which is not a retailer. This misidentification produced incorrect tool parameters (e.g., queries for 'Chase Bank'), propagating downstream errors and misaligning results with the task intent.

### B.7 OUTPUT PARSING ERRORS

**Task Query (Medium)**

As part of a recent initiative at the fictional consultancy firm BrightPath Analytics, commissioned by the renowned artist Lucia Moretti for an upcoming exhibition in Zurich, you are tasked with supporting market research on the digital art landscape. Lucia is specifically interested in public engagement with YouTube content for 'AI-generated art tools'. Retrieve the first five search results returned for this query. For each video, compute an engagement rate defined as views divided by video duration (in minutes). Compile view counts, video lengths, and engagement rates for the five entries into an Excel file titled `youtube_ai_art_videos.xlsx` for forwarding to Lucias Zurich studio.

**Agent Trajectory**

**Think:** Search YouTube for AI-generated art tools and get top-5 video IDs.
**Tool Call:**

```
{
  "name": "youtube-data.searchVideos",
  "parameters": { "query": "AI-generated art tools", "maxResults": 5 }
}
```

**Tool Call Result:** Retrieved 5 video IDs.

**Think:** Fetch detailed stats (views) and durations for these IDs.
**Tool Call:**

```
{
  "name": "youtube-data.getVideoDetails",
  "parameters": { "videoIds": ["ELr-_QbBozk","bbBJmnKH-mA","avQhPNfx-1k","SV4KSsmzv9Y","
      RL7FQcw2C6k"] }
}
```

**Tool Call Result:** Views and durations returned.

**Think:** Convert ISO durations to minutes and compute engagement (views/min) in Python.
**Tool Call:**

```
{
  "name": "code-interpreter.execute_python_code",
  "parameters": { "code": "Code to compute minutes and views/min for top-5." }
}
```

**Tool Call Result:** Computed minutes and engagement values.

**Think:** Create workbook and sheet.
**Tool Call:**

```
{
  "name": "excel.create_workbook",
  "parameters": { "filepath": "youtube_ai_art_videos.xlsx" }
}
```

**Tool Call Result:** Workbook created.

**Tool Call:**

```
{
  "name": "excel.create_worksheet",
  "parameters": { "filepath": "youtube_ai_art_videos.xlsx", "sheet_name": "Top 5 Videos" }
}
```

**Tool Call Result:** Sheet created.

**Think:** Write the views, minutes, and engagement information of the 5 videos into the Excel file.
**Tool Call:**

❌ Error

```
{
  "name": "excel.write_data_to_excel",
  "parameters": { "filepath": "youtube_ai_art_videos.xlsx",
  "sheet_name": "Top 5 Videos", "data": "The 5 videos with Views, Minutes, and
      Engagement (views/min)" }
}
```

**Tool Call Result:** First write completed. Engagement for video #2 miswritten as 81,999.9999 (should be 81,833.3333) due to an output processing error.

## Error Diagnosis

**Output parsing errors:** Due to incorrect processing of MCP tool outputs (`youtube-data.getVideoDetails`: view counts and durations), the engagement rate (views/min) was miscomputed (81,999.9999 vs. 81,833.3333).

## C  ADDITIONAL EXPERIMENTAL RESULTS

| Model | Run 1 | | Run 2 | | Run 3 | | Std. Dev. | |
|---|---|---|---|---|---|---|---|---|
| | TSR | ARS | TSR | ARS | TSR | ARS | TSR | ARS |
| GPT-5 | 58.33 | 74.25 | 56.67 | 73.92 | 58.33 | 74.08 | 0.96 | 0.17 |
| Claude-4.1-Opus | 43.33 | 62.58 | 43.33 | 62.42 | 41.67 | 61.75 | 0.96 | 0.44 |
| GPT-4.1 | 33.33 | 54.67 | 36.67 | 55.50 | 36.67 | 56.08 | 1.93 | 0.71 |
| Gemini-2.5-Pro | 28.33 | 46.17 | 30.00 | 46.42 | 26.67 | 44.92 | 1.66 | 0.81 |
| Qwen3-235B | 23.33 | 42.42 | 23.33 | 41.92 | 25.00 | 44.25 | 0.96 | 1.22 |
| Qwen3-32B | 16.67 | 32.83 | 20.00 | 34.08 | 20.00 | 34.33 | 1.92 | 0.81 |

Table 4: Stability analysis across three independent runs. Columns show performance for each run. The final columns show the Standard Deviation (SD) across runs, indicating high reproducibility.

# D PROMPTS

## D.1 RESULT EVALUATION PROMPT

You are a senior evaluator judging how well an AI agent solves a task.

Task Query: {query}.
EVALUATION INSTRUCTION: Given the query above and the reference answer, evaluate how well the agent solves the task.

LIKERT-STYLE DISCRETE SCORING (1–5)
- 5 (Excellent): Agent output conveys the same results and information as reference; task fully satisfied; differences in formatting or wording are fine
- 4 (Good): Mostly correct with minor omissions or small inaccuracies
- 3 (Fair): About half of the results is correct but some requirement not met or with noticeable inaccuracies
- 2 (Poor): Only a small portion is correct, substantially incomplete or with significant inaccuracies
- 1 (Fail): No correct or relevant results (off-topic, fabricated, or entirely incorrect)

You MUST snap to one of these exact Likert values: 1, 2, 3, 4, or 5.

CRITICAL RULES:
1. DO NOT excuse material differences due to "dynamic data" or "timing"
2. Focus on the content in both the reference and agent output that fulfills the Query's requirements and intent.
3. Structure and wording variations are acceptable
4. NUMERICAL TOLERANCE: For values that may easily fluctuate briefly (e.g., driving times, prices, view counts):
- Minor variations plausibly due to short-term fluctuation should be considered correct
- Example: "$120" vs "$118" for a specific room price is acceptable
- Example: "25 minutes drive" vs "23-27 minutes" is acceptable
- Example: "6900 view counts" vs "6908 view counts" is acceptable

Provide your evaluation in the following JSON format:

```
{
    "likert": <integer 1-5>,
    "feedback": "Detailed explanation for the chosen rating"
}
```

## D.2 TRAJECTORY EVALUATION PROMPT

You are a senior evaluator judging the overall quality of the agent's tool chain (trajectory) for solving the task.

Reference Tool Chain (for context):
{reference tool chain}

Agent's Actual Tool Chain:
{agent's actual tool chain}

LIKERT-STYLE DISCRETE SCORING (1–5):
- 5 (Excellent): The trajectory is logically sound, efficient, complete, and demonstrates strong reasoning. All necessary steps are present, no major mistakes, and the approach is either optimal or a clearly valid alternative
- 4 (Good): The trajectory is mostly correct, reasonable, and relevant; steps are generally appropriate and accurate, with noticeable but non-critical omissions or inefficiencies; no critical errors
- 3 (Fair): Some correct, relevant steps, but with gaps in logic/completeness or several questionable/inefficient choices
- 2 (Poor): Few correct steps; substantially incomplete or contains clearly wrong tool usage that undermines progress
- 1 (Failed): The trajectory does not include any correct or relevant steps toward solving the task, is illogical or largely incorrect, and does not meaningfully advance the task; not directly usable

You MUST snap to one of these exact Likert values: 1, 2, 3, 4, or 5.

Note: The agent's approach does not need to match the reference exactly. Please focus on the overall quality, efficiency, and logic of the agent's tool chain.

Provide your evaluation in the following JSON format:

```
{
    "likert": <integer 1-5>,
    "feedback": "Detailed explanation for the chosen rating"
}
```

## D.3 REACT PROMPT

> You are a helpful AI assistant. Please complete the following task:
>
> {query}
>
> Follow this structured ReAct approach:
>
> **THINK**: Analyze the current situation and choose the most appropriate tool and parameters
> **ACT**: Execute the tool call
> **OBSERVE**: Analyze results and determine next steps
>
> Repeat the process until you have completed the task. You should solve the task completely by yourself using the tools provided. Do not stop to ask for any information or guidance during the process.

## D.4 REFERENCE AGENT PROMPT

> You are a precise task executor. You must follow the execution plan exactly as specified.
>
> Task: {query}
>
> Execution plan - you must follow these steps in order:
> {execution_plan}
>
> Critical rules:
> 1. Execute each step in the exact order specified
> 2. Use the exact tool names provided
> 3. Do not skip any steps
> 4. Do not add extra steps
> 5. Do not change the order of steps
> 6. For each step, explain what you're doing and then immediately execute it
>
> Start executing step 1 now. After each tool call, verify the result and proceed to the next step.

## E LARGE LANGUAGE MODELS USAGE STATEMENT

During the preparation of this work, we used Large Language Models (LLMs) solely as a writing aid to polish the language for readability.

