# OpenReview forum: "LiveMCP-101: Stress Testing and Diagnosing MCP-enabled Agents on Challenging Queries"
_ICLR.cc/2026/Conference — Submitted to ICLR 2026_

### Official Review · Reviewer_1b2H · 2025-10-27

**Soundness:** 2
**Presentation:** 2
**Contribution:** 2
**Rating:** 2
**Confidence:** 4

**Summary:**

This paper introduces LiveMCP-101, a benchmark of 101 real-world, multi-step tasks designed to evaluate MCP-enabled agents across diverse tools (web search, file ops, math, data analysis). Key ideas are: (i) curating queries via iterative LLM rewriting + manual review; (ii) scoring against ground-truth execution plans to mitigate temporal drift; and (iii) a paired evaluation where a reference agent executes the plan while a test agent acts autonomously. The authors report <60% TSR for frontier LLMs, analyze seven failure modes, and present ablations on iteration budgets and MCP-server pool size.

**Strengths:**

- Timely topic: Evaluating live tool use under MCP is important and under active development.
- Evaluation method: Comparing an autonomous agent against a simultaneously run plan-following reference execution is a good way to reduce brittleness from time-varying tools and to enable trajectory-level diagnosis.
- Useful error taxonomy: The seven failure modes (planning, parameter, parsing) align with what many practitioners see in production agents and could guide method development.

**Weaknesses:**

1) The bench scale feels too small for the paper’s claims. 101 tasks across 41 servers / 260 tools is modest given the heterogeneity claimed. Competing datasets reach notably larger scales or cover broader API/tool surfaces. Without a power analysis, it’s hard to accept broad conclusions (e.g., token-efficiency “log-shape”) as general rather than sample-specific. Please consider expanding coverage (more domains, more servers, more tasks per domain).

2) Related-work positioning is underdeveloped and too qualitative. The narrative briefly cites MCP-specific efforts (e.g., MCP-RADAR, MCPEval) but does not systematically contrast LiveMCP-101 against:
   - MCP-centric evaluations: MCP-RADAR, MCPEval, MCP-Bench, LiveMCPBench. What exactly is novel beyond the “plan-following reference execution”? A careful table comparing #servers, #tools, #tasks, avg steps, on-/off-policy, real vs mock tools, drift handling, live or not, scoring method (LLM-judge vs references), and release artifacts is needed.
   - Non-MCP but influential agent/tool benchmarks: ToolSandbox (stateful, on-policy conversational evaluation), StableToolBench (stability via virtual APIs), API-Bank (runnable APIs + dialogues), ShortcutsBench, FAIL-TaLMs, τ-bench and τ²-bench (user-agent-tool interaction & dual-control), AssistantBench/Web agents lines. The paper’s novelty claims should be tempered and clarified against these lines.
Even if some of the works were released only recently, the community still cares deeply about seeing a comparative discussion with them.

3) Metric design.
   - The Likert-to-{0, .25, .5, .75, 1} mapping is reasonable, but the paper lacks inter-run variance, confidence intervals, and significance tests. Given live endpoints, variance across days/tools should be quantified; otherwise, model ranking may be unstable.

4) Clarity/organization issues.
   - The related-work discussion is “high-level” and doesn’t crisply delineate LiveMCP-101 from LiveMCPBench (also live MCP, multi-server) or from MCP-RADAR/MCPEval (multi-dimensional and deep evaluation). Right now, “dynamic live tools + LLM-as-judge” is not novel. Even if some of the works were released only recently, the community still cares deeply about seeing a comparative discussion with them.

5) Not open-source.

**Questions:**

1) Please solve the weakness provided.

2) Reproducibility: will you release task specs, plans, tool pools, traces, and the judge prompts?

---

> ### Author Response · Authors · 2025-11-20
> **Re Weakness 1**
>
> We agree that expanding coverage is valuable. However, we respectfully argue that the current benchmark prioritizes evaluation fidelity and diagnostic precision over raw volume, which is essential for assessing "live" agentic capabilities.
>
> **Unlike datasets relying on static offline data or purely synthetic generation** (which often contain hallucinations or outdated schemas), LiveMCP-101 operates in a real-time, dynamic environment. To ensure the reliability of evaluation in this setting, we invested about **120 expert-hours** to manually verify and refine the Execution Plans for every single query (Section 3.1).
>
> **Depth vs. Breadth**: Our tasks are multi-step, requiring an average of 5.4 tool calls (up to 15). This means our 101 tasks represent over 500 distinct reasoning and execution decision points.
>
> **Effect Size**: The performance gaps observed are substantial (e.g., GPT-5 at ~58% vs. Llama-3.1 at ~1%). Given these large effect sizes, 101 samples provide sufficient statistical power to distinguish model tiers and observe trends like the "log-shape" token efficiency, which holds consistently across multiple model families (Figure 4b).
>
> Moreover, In the LLM coding and reasoning domain, highly curated "Golden Sets" like **HumanEval (164 tasks)** have proven to be more robust estimators of model capability than larger, noisier datasets. LiveMCP-101 aims to serve a similar role as a high-standard diagnostic benchmark for MCP agents.

---

> ### Author Response · Authors · 2025-11-20
> **Re Weakness 2 & Weakness 4**
>
> To systematically delineate LiveMCP-101, we present the comparison in **Table 1**. Beyond the reference mechanism, our core novelty lies in resolving the "Dynamic Evaluation": LiveMCP-101 is the **only** benchmark that achieves verifiable ground truth within temporally dynamic, real-world environments, whereas concurrent works like LiveMCPBench lack reliable ground truth for live tools. Furthermore, we target significantly higher complexity (5.4 average steps) and cross-domain orchestration compared to the MCPEval or MCP-RADAR.
>
>
>
>
> **Table 1: Comparison of LiveMCP-101 with Related MCP Benchmarks**
> | Benchmark | # MCP servers | Avg # Tool Calling Steps | Real-World Live Integration | Temporal Dynamics | Verifiable Ground Truth | Cross-Domain Tasks|
> | :--- | :---: | :---: | :---: | :---: | :---: | :---: |
> | MCPWorld | 10 | - | ✅ | ❌ | ✅ | ❌ |
> | MCP-RADAR | 9 | - | ❌ | ❌ | ✅ | ❌ |
> | MCPEval | 19 | - | ❌ | ✅ | ❌ | ❌ |
> | LiveMCPBench (Concurrent) | 70 | 2.8 | ✅ | ✅ | ❌ | ✅ |
> | **LiveMCP-101 (Ours)** | **41** | **5.4** | ✅ | ✅ | ✅ | ✅ |
>
>
>
>
> **While non-MCP benchmarks** make important progress on tool use, as stated in **Line 62-70**, they **all** evaluate models in simulated, virtualized, or controlled environments, ranging from mock databases to virtual APIs to curated website interaction. **None** evaluate agents on live, time-varying, multi-server real-world tools.

---

> ### Author Response · Authors · 2025-11-20
> **Re Weakness 3**
>
> we conduct additional experiments on a stratified subset of tasks for six representative models: GPT-5, Claude-4.1Opus (ET), GPT-4.1, Gemini-2.5-Pro, Qwen3235B-A22B, and Qwen3-32B. We evaluate a sampled set of 60 tasks in total with 20 per difficulty tier (Easy, Medium, Hard).
>
>
> **The results (Table 2 below) demonstrate that LiveMCP-101 is highly stable, with low variance and consistent model rankings across runs.**
>
> Table 2: Inter-run Stability Analysis (3 Independent Runs)
> | Model | Run 1 TSR | Run 1 ARS | Run 2 TSR | Run 2 ARS | Run 3 TSR | Run 3 ARS | TSR Mean | TSR Variance | ARS Mean | ARS Variance |
> | :--- | :--- | :--- | :--- | :--- | :--- | :--- | :--- | :--- | :--- | :--- |
> | **GPT-5** | 58.33 | 74.25 | 56.67 | 73.92 | 58.33 | 74.08 | 57.78 | 0.612 | 74.08 | 0.018 |
> | **Claude-4.1-Opus (ET)** | 43.33 | 62.58 | 43.33 | 62.42 | 41.67 | 61.75 | 42.78 | 0.612 | 62.25 | 0.129 |
> | **GPT-4.1** | 33.33 | 54.67 | 36.67 | 55.50 | 36.67 | 56.08 | 35.56 | 2.479 | 55.42 | 0.335 |
> | **Gemini-2.5-pro** | 28.33 | 46.17 | 30.00 | 46.42 | 26.67 | 44.92 | 28.33 | 1.848 | 45.84 | 0.431 |
> | **Qwen3-235B-A22B** | 23.33 | 42.42 | 23.33 | 41.92 | 25.00 | 44.25 | 23.89 | 0.620 | 42.86 | 1.003 |
> | **Qwen3-32B** | 16.67 | 32.83 | 20.00 | 34.08 | 20.00 | 34.33 | 18.89 | 2.464 | 33.75 | 0.431 |

---

> ### Author Response · Authors · 2025-11-20
> **Re  Weakness 5 & Questions 2**
>
> **Yes. We are fully committed to ensuring the reproducibility of our research.** As stated in our Reproducibility Statement (Section 8), we commit to making our source code and benchmark publicly available upon acceptance of the paper.
>
>
> This public release will include all the components the reviewer has asked for:
> 1. The 101 task queries.
> 2. The validated execution plans.
> 3. The per-task tool pools.
> 4. The collected agent traces used in our analysis.
>
>
> Furthermore, for full transparency, we have already included the complete judge prompts for both result and trajectory evaluation under Appendix C.1 and C.2.
>
>
> **Thank you for your time and consideration! We sincerely hope that you find our responses convincing and would consider increasing the rating.**

---

> ### Comment · Reviewer_1b2H · 2025-11-25
>
> Thank you for your detailed response and the additional stability analysis. However, after carefully considering your feedback, my primary concerns remain unresolved:
>
> I am still not convinced that 101 tasks are sufficient to support broad claims about general agentic capabilities across heterogeneous domains. The argument for "fidelity over volume" does not fully mitigate the statistical limitations of such a small sample size.
>
> The distinction from existing, influential non-MCP tool-use benchmarks (e.g., ToolSandbox, StableToolBench) remains weak. The contribution appears to be primarily an implementation adaptation (using MCP) rather than a fundamental methodological innovation in evaluating dynamic environments.
>
> While the promise of a future release is noted, the current lack of accessible artifacts prevents the verification of the benchmark's complexity and implementation details during the review.
>
> Consequently, I will maintain my original rating.

---

> ### Author Response · Authors · 2025-12-01
>
> Thank you for your reply. We have **updated the pdf and uploaded our codebase** to the supplementary material to address your remaining concerns. We respectfully offer the following clarifications:
>
> ### 1. On the Sufficiency of 101 Tasks.
> We respectfully **argue** that 101 complex, multi-step tasks are sufficient for establishing agentic capabilities. In the domain of complex agentic reasoning, **"fidelity over volume"** is the established standard, distinguishing evaluation benchmarks from massive pre-training or fine-tuning datasets. For LiveMCP-101, we dedicated around **120 expert-hours** to manually verify and refine the queries and execution plans for every test case (Section 3.1).
>
> **Precedent in High-Impact Benchmarks:** Many seminal evaluation sets are of comparable size because the cost of verifying complex reasoning is high. like HumanEval and GAIA. **HumanEval [1]:** The gold standard for code generation contains only 164 tasks.  **GAIA [2]:** A leading general agent benchmark; its reliable validation set is small with 165 tasks.
>
> [1] Chen et al., "Evaluating Large Language Models Trained on Code".
>
> [2] Mialon et al., "GAIA: a benchmark for General AI Assistants".
>
> ### 2. Methodological Innovation vs. ToolSandbox/StableToolBench
> We have revised **Table 1 and Section 2** to explicitly contrast LiveMCP-101 with non-MCP benchmarks. Our contribution goes beyond merely adapting to the MCP format. **It represents a fundamental methodological innovation in solving the "Dynamic Ground Truth" problem—a critical bottleneck for evaluating agents in live, changing real-world environments.**
>
>
> ### 3. Open-source Code
> **We have uploaded the benchmark and code to the supplementary material.**
>
> We believe LiveMCP-101 fills the critical gap of evaluating agents in the live world where ground truth is dynamically changing, which existing static or simulated benchmarks do not address.
>
> We sincerely hope that our responses have addressed your remaining concerns. Please let us know if you have any further questions, we remain available to address them.

---

### Official Review · Reviewer_mauR · 2025-10-30

**Soundness:** 3
**Presentation:** 3
**Contribution:** 3
**Rating:** 6
**Confidence:** 3

**Summary:**

This paper proposes LiveMCP-101, a benchmark to fill a key gap in evaluating AI agents: while tool calling is vital for agents to interact with the real world and solve complex tasks , existing benchmarks fail to test how well agents handle multi-step tasks with diverse MCP tools in realistic, dynamic scenarios. LiveMCP-101 includes 101 curated real-world queries (refined via LLM rewriting and manual checks), all requiring coordinated use of multiple MCP tools. To deal with real-world tool responses’ temporal variability (a flaw of static ground-truth evaluation), the paper introduces a parallel framework: a reference agent runs a pre-validated plan simultaneously with the evaluated agent, generating real-time reference outputs. Experiments show even frontier LLMs have a Task Success Rate (TSR) below 60%—highlighting big challenges in multi-step MCP tool use.

**Strengths:**

1.  This paper addresses the flaws of existing MCP benchmarks (static, single-step) by proposing LiveMCP-101—the benchmark for dynamic real-world scenarios. It covers multi-step, cross-domain tasks and aligns with the practical deployment needs of agents.
2.  The parallel real-time evaluation (synchronized execution of dual agents) avoids the timeliness bias of dynamic data. The validated execution plans provide reliable evaluation anchors, making the design innovative.
3.  The experimental section encompasses 18 representative LLMs, quantifies capability disparities across models, and identifies 7 distinct failure modes—thereby providing clear guidance for model optimization. Several findings are interesting: for instance, open-source models are most severely impacted by syntax errors and unproductive reasoning; additionally, when increasing iterations from 25 to 50, performance plateaus (e.g., GPT-5 shows no improvement in TSR) occurs, showing the importance of   "planning quality" rather than "iteration count".
4.  The appendices include execution plans, Prompt templates, and other supporting materials, alongside clearly outlined experimental settings.

**Weaknesses:**

1.  Lacks verification of differences across multiple LLM judges.
2.  The long-term iteration of dynamic APIs may lead to changes in their call logic. This raises questions about whether the current dual agent verification framework (synchronized execution of reference and evaluated agents) remains feasible—since pre-validated execution plans for reference agents could become obsolete due to API changes, undermining the accuracy of real-time result alignment.

**Questions:**

please refer to weaknesses

---

> ### Author Response · Authors · 2025-11-19
> **Re Weakness 1**
>
> We sincerely appreciate your valuable time and insightful feedback! We address each of your questions in our responses below.
>
> We use an LLM-as-a-Judge precisely because our queries have **objective verifiable answers  (Line 164)**. For instance (as seen in Figure 2 and Appendix B), tasks require specific, verifiable information (e.g., GitHub issue titles , computed engagement rates , or specific Airbnb listings ).
>
>
>
>
> **Handling Formatting Differences:** An evaluated agent might **correctly** answer, "The temperature in New York is 75 degrees Fahrenheit," while the reference agent's output is simply "75°F." A simple string match would fail. We use an LLM-judge to semantically compare the content of the agent's answer against the reference answer, focusing on correctness regardless of minor formatting or wording differences (as seen in the prompts of LLM judge in Appendix C).
>
> **Supplemental Judge Robustness Check:** To further justify the soundness of this approach, we have supplemented our analysis by running the evaluation with different LLM-judges (GPT-4.1, Claude-4-Sonnet and Gemini-2.5-Pro) on 6 representative models across 101 tasks. The results, shown in the table below, demonstrate high correlation across all judges, reinforcing the stability and reliability of our evaluation framework.
> | Model | GPT-4.1 TSR | GPT-4.1 ARS | Claude-4-Sonnet TSR | Claude-4-Sonnet ARS | Gemini-2.5-Pro TSR | Gemini-2.5-Pro ARS |
> |-------|--------------------|--------------------|----------------------|----------------------|---------------------|---------------------|
> | GPT-5 | 58.42 | 73.02 | 57.43 | 73.68 | 58.42 | 74.57 |
> | Claude-4.1-Opus (ET) | 41.58 | 61.88 | 40.59 | 61.45 | 41.58 | 63.08 |
> | GPT-4.1 | 35.64 | 55.94 | 35.64 | 55.07 | 37.62 | 56.36 |
> | Gemini-2.5-pro| 27.72 | 46.78 | 27.72 | 45.12 | 28.71 | 47.25 |
> | Qwen3-235B-A22B | 22.77 | 42.57 | 20.79 | 42.41 | 22.77 | 42.76 |
> | Qwen3-32B | 18.81 | 34.41 | 18.81 | 34.91 | 19.80 | 33.62 |
>
>
>
> **We attribute this high consistency to the nature of our tasks, which rely on objective, verifiable answers. Consequently, the role of the LLM judge is limited to handling formatting differences in the responses, minimizing subjective variance.**

---

> ### Author Response · Authors · 2025-11-19
> **Re Weakness 2**
>
> Thank you for this insightful question!
>
>
> This is an inherent maintenance cost for **any benchmark** that chooses real-world realism over static, mock APIs. Our framework is not only aware of this challenge **but is designed to manage it**. As shown in our Construction framework (Figure 1), our workflow has a "Revise" loop. During the evaluation, when an API's logic evolves, our Reference Agent which strictly follows the pre-validated plan will generate an **explicit failure**, which triggers this revision process. Therefore, our framework remains feasible because its "validated plans" are not static but are maintainable components designed to be updated. **We commit to maintain these execution plans in future to ensure the benchmark's long-term accuracy.**
>
>
> **We sincerely hope that our responses have addressed your remaining concerns. Please let us know if you have any further questions, we remain available to address them.**

---

### Official Review · Reviewer_1kBR · 2025-10-31

**Soundness:** 3
**Presentation:** 4
**Contribution:** 3
**Rating:** 6
**Confidence:** 3

**Summary:**

This paper introduced the LiveMCP-101 dataset for benchmarking the tool-use capability of LLM agents. The LiveMCP-101 dataset comprises 101 synthetic task queries, accompanied by task-specific MCP tool pools and human-revised execution plans as reference trajectories. To obtain real-time, correct answers for time-sensitive tasks during the evaluation, the author designed a parallel evaluation mechanism, where a reference agent executes the groundtruth plans to produce real-time references for evaluating the tested models. An extensive evaluation across 18 proprietary and open-sourced LLMs shows that frontier models fail to achieve a success rate over 60%.

**Strengths:**

1. The new benchmark offers more challenging test tasks, as evidenced by longer average tool-calling steps required and lower success rates by mainstream LLMs. Given the rapid development of LLMs and the quick saturation of evaluations, this challenging new benchmark is valuable for advancing research on agentic LLMs.

2. The parallel evaluation framework provides a practical assessment for time-sensitive tasks.

3. The paper is overall well written. The core ideas and evaluation details are well presented.

**Weaknesses:**

1. Task distribution and quality are crucial for agent evaluation benchmarks. As described in 3.1, LiveMCP-101 uses queries generated by OpenAI o3 model, but the details of the generation process remain unclear (e.g., workflows and key prompts). And using synthetic task queries may raise concerns, as these test cases may deviate from real user needs or be biased towards the LLM used for synthesis. Given that existing agent benchmarks like GAIA and SWEBench provide test tasks from real people, can you elaborate on the necessity of using generated queries?

2. LLM-as-a-judge, although widely used in LLM evaluation, may still bring risks like inconsistent evaluation results and preference leakage. The author may consider further analyzing the correlation between the tested model and the judge model, and adding rule-based metrics as a supplement.

**Questions:**

See weakness.

---

> ### Author Response · Authors · 2025-11-19
> **Re Weekness 1**
>
> Thank you for your question.
>
> We did detail the generation workflow in Section 3.1, and we will gladly add the generation prompts of OpenAI o3 to the appendix in our revised manuscript.
>
> We agree that **purely synthetic, unverified queries** would be a significant flaw. However, our methodology is the **opposite** of this: we employ a rigorous human-in-the-loop (HIL) refinement process precisely to prevent such issues and ensure **high-quality, challenging, and verifiable tasks**. As detailed in Section 3.1, the LLM (OpenAI o3) is used only for generating initial query drafts. We explicitly state that these initial drafts are often flawed, "not solvable with the provided tools," or "not easily verifiable". To ensure rigor and solve this exact problem, we applied 'multiple rounds of human-in-the-loop revision'. This rigorous process is precisely how we guarantee the final queries possess **'clarity, balanced difficulty, solvability with the given tools, and objective verifiability'**(Line 163), while simultaneously ensuring they align with real user needs."
>
> **In summary, our queries are not "synthetic" in the way the reviewer implies. They are human-curated and rigorously validated tasks. Therefore, our benchmark aligns well with real user needs and will not be biased towards the LLM used for synthesis.**

---

> ### Author Response · Authors · 2025-11-19
> **Re Weekness 2**
>
> We use an LLM-as-a-Judge precisely because our queries have **objective verifiable answers  (Line 164)**. For instance (as seen in Figure 2 and Appendix B), tasks require specific, verifiable information (e.g., GitHub issue titles , computed engagement rates , or specific Airbnb listings ).
>
>
>
>
> **Handling Formatting Differences:** An evaluated agent might **correctly** answer, "The temperature in New York is 75 degrees Fahrenheit," while the reference agent's output is simply "75°F." A simple string match would fail. We use an LLM-judge to semantically compare the content of the agent's answer against the reference answer, focusing on correctness regardless of minor formatting or wording differences (as seen in the prompts of LLM judge in Appendix C).
>
> **Supplemental Judge Robustness Check:** To further justify the soundness of this approach, we have supplemented our analysis by running the evaluation with different LLM-judges (GPT-4.1, Claude-4-Sonnet and Gemini-2.5-Pro) on 6 representative models across 101 tasks. The results, shown in the table below, demonstrate high correlation across all judges, reinforcing the stability and reliability of our evaluation framework.
> | Model | GPT-4.1 TSR | GPT-4.1 ARS | Claude-4-Sonnet TSR | Claude-4-Sonnet ARS | Gemini-2.5-Pro TSR | Gemini-2.5-Pro ARS |
> |-------|--------------------|--------------------|----------------------|----------------------|---------------------|---------------------|
> | GPT-5 | 58.42 | 73.02 | 57.43 | 73.68 | 58.42 | 74.57 |
> | Claude-4.1-Opus (ET) | 41.58 | 61.88 | 40.59 | 61.45 | 41.58 | 63.08 |
> | GPT-4.1 | 35.64 | 55.94 | 35.64 | 55.07 | 37.62 | 56.36 |
> | Gemini-2.5-pro| 27.72 | 46.78 | 27.72 | 45.12 | 28.71 | 47.25 |
> | Qwen3-235B-A22B | 22.77 | 42.57 | 20.79 | 42.41 | 22.77 | 42.76 |
> | Qwen3-32B | 18.81 | 34.41 | 18.81 | 34.91 | 19.80 | 33.62 |
>
>
>
> **We attribute this high consistency to the nature of our tasks, which rely on objective, verifiable answers. Consequently, the role of the LLM judge is limited to handling formatting differences in the responses, minimizing subjective variance.**
>
> **We sincerely hope that our responses have addressed your remaining concerns. Please let us know if you have any further questions, we remain available to address them.**

---

### Official Review · Reviewer_3NZi · 2025-10-31

**Soundness:** 2
**Presentation:** 4
**Contribution:** 2
**Rating:** 2
**Confidence:** 4

**Summary:**

This paper proposes LiveMCP-101, a benchmark of 101 multi-step real-world tasks requiring coordinated use of multiple tools via the Model Context Protocol (MCP). A parallel reference-agent evaluation is also introduced to handle dynamic tool outputs.

**Strengths:**

1. Massive MCP servers and tools are considered, rendering the benchmark comprehensive.
2. A new evaluation framework is proposed to handle dynamic tool outputs.
3. Extensive experiments are conducted to benchmark LLMs.

**Weaknesses:**

1. **Weak Motivation:** From the perspective of LLMs, there is no apparent difference between using MCP and function calling. In this regard, I am not convinced about the motivation of this paper. Why do we need to use MCP to benchmark LLMs given that massive function-calling benchmarks already exist? For example, the latest BFCL V4 benchmark already covers multi-turn and complex tool usage scenarios. The authors are recommended to clarify the motivation of this paper. Note that the cited works on MCP benchmarking in section 2 have not been accepted by any top-tier conferences, which cannot strongly support the motivation of this paper.
2. **Soundness of the Evaluation Framework:** The proposed parallel reference-agent evaluation framework is interesting. However, I have some concerns about its soundness. Specifically, the reference agent is designed to strictly follow the ground-truth solution steps, which may not reflect the actual behavior of real-world agents. Besides, the final evaluation is based on an LLM-based judge, which always suffers from LLM's non-determinism and instability issues. Given that benchmarking is a critical task in the research of LLMs, the soundness of the evaluation framework is of great importance. The authors are recommended to further justify the soundness of their proposed evaluation framework.
3. **Unclear Difficulty Levels:** It is commendable that the authors design three difficulty levels to enable fine-grained analysis. However, the criteria for difficulty level design are not clearly illustrated in the paper. For example, what is the difference between "easy" and "hard" tasks? For different LLMs, does there exist a discrepancy in task difficulty levels?

**Questions:**

1. What is the motivation of using MCP to benchmark LLMs given that massive function-calling benchmarks already exist?
2. How to ensure the soundness of the proposed evaluation framework?
3. What are the criteria for difficulty level design?

---

> ### Author Response · Authors · 2025-11-19
> **Re Weaknesses 1 &  Questions 1 on Motivation**
>
> We sincerely appreciate your valuable time and feedback! We address each of your questions in our responses below.
>
> 1. **Existing tool-calling benchmarks do not interact with the real world comprehensively.** As stated in Lines 64–66 of our paper, current tool-calling benchmarks primarily rely on synthetic or static environments. Even very recent work such as BFCL v4 conducts most of its evaluation within custom API codebases and emulated service functions, and its only live web-search component is explicitly restricted to “recent but stable” facts to avoid real-world dynamism. **In contrast**, MCP enables agents to interact with a wide range of live, real-world services under a unified protocol. Our work is specifically designed to evaluate model performance under these genuinely dynamic MCP service conditions.
>
>
> 2. **Regarding your observation that the MCP-related papers we cite have not been accepted by top-tier conferences:** We agree that this was largely accurate when the field first emerged. However, research on MCP-based evaluation is progressing rapidly. In fact, one recent MCP-focused study (**MCPEval**, https://aclanthology.org/2025.emnlp-demos.27/) has already been accepted to **EMNLP 2025**, reflecting growing recognition of the importance of this line of work. We view this not as a weakness, but as evidence that MCP-enabled agent evaluation is an early-stage, fast-developing area that still lacks well-established benchmarks. As noted in Line 126 of our paper, “MCP has been quickly adopted across all major AI players,” which further underscores the urgent need for rigorous evaluation frameworks tailored to real-world MCP services.

---

> ### Author Response · Authors · 2025-11-19
> **Re Weaknesses 2 &  Questions 2 on Soundness of the Evaluation Framework**
>
> Thank you for your **positive feedback** on our "parallel reference-agent evaluation framework" and for raising crucial questions regarding its soundness. We appreciate the opportunity to clarify our methodology and further justify its robustness.
>
>
> ### 1. The Role of the Reference Agent and Execution Plan
> We wish to clarify a key aspect of our design: the reference agent's purpose is not to define the only correct execution path, but to generate the dynamic, real-time ground-truth answer.
>
>
> **Dynamic Environments Require Dynamic Ground Truth:** As our benchmark, LiveMCP-101, evaluates agents on real-world tasks using live MCP, the correct answers are inherently dynamic and time-varying. A static ground-truth answer (e.g., a specific weather temperature) would be incorrect several hours later.
>
>
> **Objective Verifiability:** We carefully designed our 101 queries to be objectively verifiable, avoiding subjective questions (Line 164). For instance (as seen in Figure 2 and Appendix B), tasks require specific, verifiable information (e.g., GitHub issue titles , computed engagement rates , or specific Airbnb listings ).
>
>
> **Generating the "Live" Ground Truth:** The sole purpose of the reference agent, which strictly follows its validated execution plan , is to execute a known-correct sequence of operations to produce the correct, real-time answer at the moment of evaluation. For example, for a query "What is the current weather in New York?", the reference agent's plan guarantees it calls the correct tool (get_weather) with the correct parameters (city="New York"), thereby fetching the objectively correct temperature at that exact time. This "Real-Time Ref" becomes the ground truth against which the evaluated agent's final answer is compared.
>
>
> ### 2. Justification of the LLM-as-a-Judge
> We use an LLM-as-a-Judge precisely because our queries have **objective verifiable answers**, but the format of the agent's response may vary.
>
>
> **Handling Formatting Differences:** An evaluated agent might **correctly** answer, "The temperature in New York is 75 degrees Fahrenheit," while the reference agent's output is simply "75°F." A simple string match would fail. We use an LLM-judge to semantically compare the content of the agent's answer against the reference answer, focusing on correctness regardless of minor formatting or wording differences (as seen in the prompts of LLM judge in Appendix C).
>
>
> **Validated Human-Alignment:** We did not simply assume the judge's reliability. As detailed in Section 4.4, we conducted a human-expert study to assess reliability. Our results show a high, consistent agreement between the LLM-judge and human raters (quadratic-weighted Cohen’s $\kappa$ > 0.85 for result evaluation), confirming that our judge is stable and human-aligned.
>
>
> **Supplemental Judge Robustness Check:** To further justify the soundness of this approach, we have supplemented our analysis by running the evaluation with different LLM-judges (GPT-4.1, Claude-4-Sonnet and Gemini-2.5-Pro) on 6 representative models across 101 tasks. The results, shown in the table below, demonstrate high correlation across all judges, reinforcing the stability and reliability of our evaluation framework.
>
>
> | Model | GPT-4.1 TSR | GPT-4.1 ARS | Claude-4-Sonnet TSR | Claude-4-Sonnet ARS | Gemini-2.5-Pro TSR | Gemini-2.5-Pro ARS |
> |-------|--------------------|--------------------|----------------------|----------------------|---------------------|---------------------|
> | GPT-5 | 58.42 | 73.02 | 57.43 | 73.68 | 58.42 | 74.57 |
> | Claude-4.1-Opus (ET) | 41.58 | 61.88 | 40.59 | 61.45 | 41.58 | 63.08 |
> | GPT-4.1 | 35.64 | 55.94 | 35.64 | 55.07 | 37.62 | 56.36 |
> | Gemini-2.5-pro | 27.72 | 46.78 | 27.72 | 45.12 | 28.71 | 47.25 |
> | Qwen3-235B-A22B | 22.77 | 42.57 | 20.79 | 42.41 | 22.77 | 42.76 |
> | Qwen3-32B | 18.81 | 34.41 | 18.81 | 34.91 | 19.80 | 33.62 |
>
>
> **We attribute this high consistency to the nature of our tasks, which rely on objective, verifiable answers. Consequently, the role of the LLM judge is limited to handling formatting differences in the responses, minimizing subjective variance.**
>
>
> **We have clarified the soundness of our evaluation framework in our revised manuscript.**

---

> ### Author Response · Authors · 2025-11-19
> **Re Weaknesses 3 &  Questions 3 on Difficulty Levels**
>
> Thank you for **commending** our design of three difficulty levels for fine-grained analysis.
> ### 1. Difficulty Tier Annotation Methodology
> The difficulty tiers were determined through **independent assessments, voting, and discussion among the three authors**. Each author first rated the difficulty of every task individually, after which the group conducted a vote and, when necessary, a discussion to reach consensus. The final ‘Easy,’ ‘Medium,’ or ‘Hard’ assignments were based on two primary dimensions:
>
> (1) Complexity of Reasoning (tool planning and parameterization)
>
> (2) Tool Chain Length (the tool chain length in the validated execution plan).
>
> We concede that the paper did not explicitly detail the formal methodology for this stratification, and we will add this clarification to our revised manuscript.
>
> ### 2. What is the difference between 'easy' and 'hard' tasks?
>
> **Easy tasks (e.g., Figure 2, Easy )** require minimal reasoning and have a short tool chain. For instance, the Easy example simply requires fetching issues from a known repository and writing them to a file, a straightforward 2-step process (see Appendix A, Figure 8 ).
>
> **Hard tasks (e.g., Figure 2, Hard )** require complex reasoning and a long, multi-step tool chain. The Hard example demands that the agent first solve a riddle to identify the correct NBA team, handle dynamic time, process multiple complex constraints (budget, walking distance/speed), and synthesize data from several tools (time, Airbnb, Google Maps, code interpreter) . This complex logic is reflected in its 6-step execution plan (see Appendix A, Figure 10 ).
>
> The reviewer rightly asks if this difficulty is objective. Our results in Table 1 in our paper empirically validate our methodology. As shown, performance consistently degrades across all models as task difficulty increases (e.g., GPT-5 drops from 86.67% on Easy to 39.02% on Hard; Qwen3-235B-A22B drops from 43.33% to 4.88% ). This confirms our stratification captures objective, inherent task difficulty, independent of any single model.

---

> > ### Comment · Reviewer_3NZi · 2025-11-21
> >
> > Thanks for your reply. I found the responses to my second and third questions to be satisfactory, so I would like to increase the rating from 2 to 4. However, regarding the first question, I still think the response is not quite convincing. The response states that the tool-calling benchmark mainly relies on synthetic data, but this paper also employs LLMs to generate the test cases.
> >
> > Furthermore, the goal of this paper and previous tool-calling benchmarks are the same, i.e., to evaluate the capability of the LLM agent. In this regard, when we discuss the contribution of a new benchmark, we would primarily assess its contribution in terms of this goal. However, to the best of my knowledge, the difference between tool-calling and MCP is the implementation of external tool invocation. In other words, tool-calling and MCP are equivalent from the perspective of LLM agents. Then what is the motivation for employing MCP instead of tool-calling? And more importantly, what is the core contribution of this paper compared with previous tool-calling benchmarks? The authors are highly recommended to clarify these questions with solid evidence.

---

> ### Author Response · Authors · 2025-11-21
>
> We sincerely thank you for finding **our responses to the evaluation framework and difficulty levels satisfactory**. We deeply appreciate the opportunity to clarify the motivation and the critical distinction between previous tool-calling benchmarks and our LiveMCP-101.
>
> ### 1. Clarification on "Synthetic Data": Synthetic Queries vs. Real-World Environments
> We respectfully point out that there is a **fundamental distinction** between how a query is generated and the environment in which it is executed.
>
> **Previous Benchmarks rely on Synthetic Environments:** Existing benchmarks (e.g., ToolBench, ToolACE, BFCL) rely on synthetic tools and mock, static databases. In these setups, the agent interacts with a simulator that returns deterministic outputs. This fails to capture the complexity and dynamism of real-world scenarios.
>
> **LiveMCP-101 operates in Real-World Environments:** While we utilize LLMs to assist in drafting the initial queries, the execution environment and the answers are 100% real and live. For example, When an agent in our setting calls github.search_issues, it hits the actual GitHub API, returning real and live data. When it calls weather.get, it fetches the actual current weather.
>
> **Human-Verified Queries:** Furthermore, our queries are not raw LLM synthetic data. As detailed in Section 3.1, we employ a rigorous Human-in-the-Loop revision process (~120 expert hours) to to ensure task quality. Therefore, our benchmark aligns well with real user needs and will not be biased towards the LLM used for synthesis.
>
> ### 2. Motivation for MCP Benchmark: Aligning Evaluation with Emerging Standards
> **The MCP is rapidly emerging as a critical standard and has been quickly adopted across all major AI players (e.g., Anthropic, OpenAI, Google)**. As noted above, existing tool-calling benchmarks typically constrained to synthetic environments, which **cannot** faithfully reflect an agent's ability to operate in real, dynamic environments. Therefore, a Live MCP benchmark is essential to rigorously stress-test agents against the specific challenges of this widely adopted industrial standard , filling a significant gap in evaluating how agents handle the complexity and heterogeneity of real-world MCP servers.
>
> **The Difference between MCP and traditional tool calling:** Prior tool calling setups **do not** support heterogeneous multi-server environments, live real-world APIs, or temporally varying tool responses. MCP is the first widely adopted protocol that exposes such a dynamic, multi-server system. Consequently, the capabilities required of agents, such as cross-server planning and handling non-deterministic outputs, are fundamentally different from those evaluated in existing tool-calling benchmarks.
>
> **Prevention of Data Contamination:** Static benchmarks rely on fixed ground truth answers. Since these answers are static, there is a significant risk that they are included in the pre-training process of frontier models, leading to data leakage where models "memorize" rather than "solve." In contrast, LiveMCP-101 tasks depend on dynamic, time-varying states. The correct answer changes over time, making memorization impossible and forcing the agent to demonstrate genuine tool-use capabilities.
>
> ### 3. Core Contributions of Our Paper
> To summarize, our core contributions that advance the field beyond existing benchmarks are:
>
> **Shift from Static to Live Evaluation:** We move the benchmark standard from synthetic, static environments to dynamic, real-world environments. This tests robustness and eliminates data leakage risks inherent in static datasets.
>
> **Methodological Contribution (The Parallel Reference Framework):** We solve the open problem of "How to evaluate agents in dynamic environments?" Previous benchmarks avoided live web/API tasks and used mock databases because the "ground truth" changes (e.g., the weather of New York). Our Parallel Reference Agent approach, anchored in validated execution plans , allows the community to evaluate agents in dynamic environments by producing real-time reference outputs.
>
> **Diagnosing "Real-World" Failures:** Because we use live MCP tools, we uncovered failure modes in real, live environments that static benchmarks miss. We provide a detailed failure analysis on representative models , identifying three primary error categories: tool planning and orchestration errors, parameter errors, and output handling errors with seven subtypes. This analysis informs targeted improvements to enhance agent capability in dynamic, multi-step MCP tool orchestration.
>
> **Thank you for your time and consideration! We sincerely hope that our responses have addressed your remaining concerns. Please let us know if you have any further questions, we remain available to address them.**

---

> > ### Comment · Reviewer_3NZi · 2025-11-23
> >
> > Well, I believe we have had a fruitful discussion, and I have a better understanding of this paper now. Thanks for your reply. To summarize, the core contribution of this paper is to propose a new benchmark that introduces a dynamic environment with a tailored test oracle (based on a "parallel reference-agent"). In this regard, I have the following suggestions for the authors to revise their paper:
> >
> > 1. Add a brief discussion about the difference between the function-calling framework and MCP, especially from the perspective of **the implementation of the "dynamic environment,"** in order to justify the selection of MCP and clarify the contribution of this paper. Note that the function-calling framework, indeed, is also capable of implementing a "dynamic environment", so it would be helpful if the authors can clarify why they choose MCP. In addition, regarding the "data contamination," I am not quite convinced by the response. For most of the "dynamic" test cases, e.g., temperature query, though the values may vary over time, the general action traces may still be similar. Is there any more solid evidence to support this claim?
> >
> > 1. Add more analysis about the effectiveness of the proposed new benchmarking methodology. This is very important. In terms of diversity, difficulty, real-world representativeness, and other aspects, how does the proposed benchmark compare with previous tool-calling benchmarks? Do the "dynamic environment" and "parallel reference-agent" really help to better evaluate the capability of LLM agents? Is there any quantitative or qualitative evidence to support this claim? It is highly recommended to design several **comparative experiments** to illustrate the superiority of the proposed benchmark over previous ones.
> >
> > Considering the above suggestions would require a complete revision of the paper, I would like to keep my overall rating as 4. However, I would not mind if this paper got accepted. I hope my comments and suggestions could help the authors to improve their work.

---

> ### Author Response · Authors · 2025-11-26
>
> We sincerely thank you for the **fruitful discussion** and your **constructive suggestions** to strengthen our work. **We are greatly encouraged by your positive feedback and your openness to supporting its acceptance after the revision.**
>
> We have **carefully revised the paper** (highlighted in blue) to incorporate **all your suggestions** regarding the justification of MCP vs. tool calling, the soundness of the evaluation framework, the criteria for task difficulty, and the effectiveness of our methodology.
>
> ### 1. Why choose MCP if traditional function-calling can also implement dynamic environments?
>
> We have **revised Section 2 (Related Work)** to clarify this distinction. We split the "Related Work" into "Agents with Tool Use" and "Model Context Protocol (MCP)" to explicitly contrast the two. We also added **Table 1**, which systematically compares LiveMCP-101 with prior benchmarks.
>
>
> ### 2. Effectiveness of the Proposed Benchmarking Methodology
>
> Thank you for highlighting this critical aspect. We do not claim that dynamic evaluation is inherently "better" than static evaluation for all purposes; rather, **they represent distinct evaluation tracks serving different dimensions.**
>
> Evaluating MCP-enabled agents in a static, mocked environment is similar to testing autonomous vehicles solely in simulators. While simulators are valuable for initial testing, they cannot replicate the "Sim-to-Real" gap: the physical unpredictability, latency, and noise of real roads. Similarly, evaluating robotic manipulation often requires moving from simulation to physical execution to handle real-world friction and sensor noise.
>
> To demonstrate that our "Parallel Reference-Agent" framework produces reliable and valid evaluations despite the dynamic nature of the environment, we provide the following quantitative evidence **(incorporating new experimental results to our paper):**
>
> **High Alignment with Human Experts**: As shown in Figure 6, the agreement between our LLM Judge and human experts is high (Cohen’s $\kappa > 0.85$). This quantitatively proves that our Reference-Agent-based scoring aligns with human judgment on correctness.
>
> **Judge-Judge Consistency (New Experiment)**: To ensure the evaluation is not biased by a specific judge model, we conducted a new cross-validation experiment using three different LLMs (GPT-4.1, Claude-4-Sonnet, Gemini-2.5-Pro) as judges. These results have been added to the paper in the new Table 3, showing minimal standard deviation across judges.
>
> **Run-to-Run Stability (New Experiment)**: We performed stability analysis by running the evaluation three independent times. As shown in the new Table 4 (Appendix C), the performance variance is minimal, demonstrating the robustness of our parallel reference strategy.
>
>
> **We have implemented these changes in the respective sections of the paper. Please let us know if you have any further questions, we remain available to address them.**
>
> **Once again, we sincerely appreciate your willingness to support the acceptance of our paper after the revision.**

---

### Author Response · Authors · 2025-12-01
**Summary of Revisions and Global Response**

We are grateful to the reviewers for their constructive feedback and suggestions, which have helped us to improve our work.

In this paper, we introduce **LiveMCP-101**, a benchmark designed to test the multi-step reasoning and tool orchestration capabilities of agents within the **dynamic Model Context Protocol (MCP) system**. By establishing a novel parallel evaluation framework that leverages validated execution plans to mitigate the temporal variability of real-time data, LiveMCP-101 exposes critical reliability gaps in even frontier LLMs and provides a detailed taxonomy of failure modes spanning tool planning, parameterization, and output handling to guide the development of the agentic systems.

We are greatly encouraged by the reviewers’ **positive feedback on our motivation and the significance of this work.** Specifically, reviewers highlighted:

**1. Timely and Important Research Topic:**
- Evaluating live tool use under MCP is important and under active development. (Reviewer 1b2H)

- This challenging new benchmark is valuable for advancing research on agentic LLMs. (Reviewer 1kBR)

- Aligns with the practical deployment needs of agents. (Reviewer mauR)

**2. Addressing Critical Gaps in Existing Benchmarks:**
- Addresses the flaws of existing MCP benchmarks (static, single-step) and covers multi-step, cross-domain tasks. (Reviewer mauR)
- The new benchmark offers more challenging test tasks. (Reviewer 1kBR)
- Massive MCP servers and tools are considered, rendering the benchmark comprehensive. (Reviewer 3NZi)

**3. Innovative Evaluation Framework:**
- A new evaluation framework is proposed to handle dynamic MCP outputs. (Reviewer 3NZi)
- Comparing the evaluated agent against a simultaneously run plan-following reference execution is a good way to reduce brittleness.(Reviewer 1b2H)
- The parallel evaluation framework provides a practical assessment for time-sensitive tasks. (Reviewer 1kBR)

Reviewers provided many insightful and constructive suggestions. We have endeavored to address all concerns by adding detailed descriptions and required results. Below is a summary of the primary modifications:

**1. Clarification of Motivation and Comparison with Related Work** (Reviewer 3NZi, Reviewer 1b2H): To address concerns regarding the necessity of MCP benchmarking and novelty, we expanded the introduction and related work sections. We explicitly contrasted LiveMCP-101 with static function-calling benchmarks and concurrent MCP works, highlighting our focus on multi-step reasoning with verifiable ground truth in dynamic environments. Refer to Section 1, Section 2, and Table 1 in the revised version.

**2. Validation of LLM-as-a-Judge Reliability and Consistency** (Reviewer 3NZi, Reviewer 1kBR, Reviewer mauR): Following recommendations to ensure evaluation soundness, we conducted cross-validation using three different LLM judges (GPT-4.1, Claude-4-Sonnet, Gemini-2.5-Pro). Refer to Section 4.4 and Table 3 in the revised version.

**3. Stability Analysis and Experimental Variance** (Reviewer 1b2H, Reviewer mauR): To address concerns about the instability of live evaluations, we performed stability analysis across three independent runs. We reported the Standard Deviation for Task Success Rates (TSR) and Average Result Scores (ARS), confirming the stability of our results despite real-time data fluctuations. Refer to Table 4 in the revised version.

**4. Clarification on Task Construction and Difficulty Criteria** (Reviewer 3NZi, Reviewer 1kBR): We refined the description of our dataset construction to address concerns about synthetic queries and unclear difficulty levels. We explicitly defined the three difficulty tiers based on tool chain length and reasoning complexity, and detailed the human-in-the-loop revision process used to ensure queries reflect realistic user needs. Refer to Section 3.1 in the revised version. We have also uploaded the benchmark and source code to the supplementary material.


We once again thank the Reviewers, Area Chairs, and Program Chairs for their time and dedication to the review process.

---

### Meta-Review · Area_Chair_Aq6X · 2026-01-04

**Summary:**

The decision to reject is primarily driven by concerns regarding the scale and scope of the proposed benchmark. While the parallel reference-agent evaluation framework was appreciated, multiple reviewers felt that 101 tasks are insufficient to support broad claims about general agentic capabilities. Furthermore, the motivation for a strictly MCP-specific benchmark, as opposed to broader tool-use evaluations, was not convincingly established to differentiate it from existing work.

**Reviewer Concerns:**

The authors successfully addressed concerns regarding the reliability of the LLM-as-a-judge framework and the criteria for task difficulty stratification by providing additional stability analysis and validation data. However, the core concerns regarding the small sample size and the limited novelty compared to established non-MCP tool-use benchmarks remain outstanding and were the primary factor in the final decision.

**Reviewer Scores:**

Reviewer 3NZi explicitly raised their score to 4 during the discussion but remained hesitant; they might have considered a weak acceptance if the motivation was clearer. Reviewers 1kBR and mauR likely maintained their borderline positive scores 6 as their specific concerns were mostly technical. However, Reviewer 1b2H explicitly stated they would maintain their low score 2, as the fundamental issue of benchmark scale was not resolved by the rebuttal.

---

### Decision · Program_Chairs · 2026-01-26

Reject